# Rapid assay development for low input targeted proteomics using a versatile linear ion trap

Ariana E. Shannon [1,2], Rachael N. Teodorescu [1], No Joon Song[1], Lilian R. Heil[3], Cristina C. Jacob[3], Philip M. Remes[3], Zihai Li [1], Mark P. Rubinstein[1,4] & Brian C. Searle [1,2] ✉

Advances in proteomics and mass spectrometry enable the study of limited cell populations, where high-mass accuracy instruments are typically required. While triple quadrupoles offer fast and sensitive low-mass specificity measurements, these instruments are effectively restricted to targeted proteomics. Linear ion traps (LITs) offer a versatile, cost-effective alternative capable of both targeted and global proteomics. Here, we describe a workflow using a hybrid quadrupole-LIT instrument that rapidly develops targeted proteomics assays from global data-independent acquisition (DIA) measurements without high-mass accuracy. Using an automated software approach for scheduling parallel reaction monitoring assays (PRM), we show consistent quantification across three orders of magnitude in a matched-matrix background. We demonstrate measuring low-level proteins such as transcription factors and cytokines with quantitative linearity below two orders of magnitude in a 1 ng background proteome without requiring stable isotope-labeled standards. From a 1 ng sample, we found clear consistency between proteins in subsets of CD4+ and CD8+ T cells measured using high dimensional flow cytometry and LIT-based proteomics. Based on these results, we believe hybrid quadrupole-LIT instruments represent a valuable solution to expanding mass spectrometry in a wide variety of laboratory settings.

Systems biology is the study of interactions within and between cells, where the goal is to learn how those interactions give rise to the complex behavior seen in an entire system[1]. One challenge is that many complex biological processes, such as adaptive immunity, are built from small populations of distinct cell types acting in concert[2,3]. Improvements in proteomics methods and mass spectrometry (MS) instrumentation have paved the way for low-input and single-cell proteomics, which make it possible to study how limited cell populations contribute to the whole. While the majority of single-cell

methods use tandem mass tags (TMT)[4] to increase signal (and thus consistency) with data-dependent acquisition (DDA)[5,6], several groups have demonstrated that data-independent acquisition (DIA) is an effective solution to measuring low-input samples[7–9]. However, high-mass accuracy instruments are used in nearly all cases.

While single-cell and low-input global proteomics is typically acquired using high-mass accuracy instruments, nominal-mass instruments, such as triple quadrupoles, lead in quantitative sensitivity using targeted selected reaction monitoring (SRM)[10]. With SRM,

[1]Pelotonia Institute for Immuno-Oncology, The Ohio State University Comprehensive Cancer Center, Columbus, OH 43210, USA. [2]Department of Biomedical Informatics, The Ohio State University Medical Center, Columbus, OH 43210, USA. [3]Thermo Fisher Scientific, San Jose, CA 95134, USA. [4]Division of Medical Oncology, Department of Internal Medicine, The Ohio State University College of Medicine, Columbus, OH 43210, USA. ✉e-mail: brian.searle@osumc.edu; searle.brian@mayo.edu

peptides are detected based on monitoring multiple fragment ion signals produced by each selected precursor ion. Transitions (diagnostic precursor/fragment ion pairs) in a pre-specified schedule must be provided to the instrument for monitoring at specific times within the chromatographic gradient[11]. While triple quadrupoles are extremely capable instruments that can rapidly switch between ion pairs (typically 0.5 msec dwell time), they can only monitor a single $m/z$ at a time, which makes generating even low-resolution full spectra impractical. As such, triple quadrupoles are practically limited to targeted experiments, which traditionally require a high-mass resolution instrument to select and schedule targeted peptides and transitions before migrating to a nominal-mass instrument for high-throughput monitoring.

An alternative targeted method to SRM is parallel reaction monitoring (PRM), which uses a quadrupole-equipped high-resolution mass spectrometer where the third quadrupole is replaced with an Orbitrap™ (Q-Orbitrap, also known as a Q-Exactive™) or a time-of-flight analyzer (Q-ToF). Rather than measure precursor/fragment transitions, all precursor-specific fragment ions are collected in a full tandem mass spectrum with PRM[12]. A major advantage of PRM is that diagnostic fragment ions are selected after the experiment is performed, which can vastly simplify the assay development process. PRM has provided meaningful biological insight into several diseases, including systemic autoimmune diseases[13], multiple sclerosis[14], and colorectal cancer[15]. When coupled with global proteomics, PRM is a powerful tool for interrogating system-wide interactions between cells.

Linear ion traps (LITs) are another versatile, fast, nominal-mass analyzer comparable in resolution and complexity to triple quadrupoles. Modern Thermo Scientific™ Tribrid™ instruments have incorporated LITs as a tertiary analyzer, coupled with an Orbitrap[16]. Using a Tribrid instrument, Heil et al.[17] showed that the benefit of PRM lies within its ability to monitor multiple product ions produced within a selected precursor $m/z$ range and that the LIT in Tribrids was an effective readout for targeted proteomics. A LIT measures ions trapped in an electric field by adjusting RF and DC voltages to selectively eject ions based on their $m/z$ to collect MS/MS spectra. Unlike triple quadrupoles, which have to "dwell" at each increment of $m/z$ to form a spectrum, LITs acquire full scan MSn data quickly and sensitively[18], making them viable for global proteomics using DDA or DIA[19]. As a result, a hybrid quadrupole-LIT (Q-LIT) could act as an "all-in-one" nominal-mass instrument capable of both targeted and global proteomics.

As with triple quadrupoles, LITs are extremely sensitive, ion-efficient mass analyzers apt for low-input proteomics, as defined by ≤100 ng per injection[20]. In some circumstances, LITs can be more effective than high-resolution mass analyzers for low-input samples, specifically at ≤10 ng[21], and can measure single cells without multiplexing reagents[22]. At higher sample input (≥100 ng), the lack of high mass accuracy overshadows the increased sensitivity of LITs. There exist other compelling reasons to consider LIT-based instruments in high-throughput applications. In particular, LITs operate at high pressure ($10^{-3}$ mTorr) in comparison to ToF analyzers ($10^{-6}$ mTorr), where ions have to travel uninterrupted for meters, or Orbitrap analyzers ($10^{-10}$ mTorr), where ions can travel for more than a kilometer. Lower vacuum pump requirements allow LITs to be built with simpler vacuum requirements than higher resolution instruments, such as Orbitrap or ToF analyzers, and housed in smaller instrument footprints.

Here, we present a workflow using a hybrid quadrupole-LIT (Q-LIT) instrument from Thermo Scientific as a single instrument for rapidly generating targeted assays for low-input experiments. With the Q-LIT, we demonstrate how to build nominal-mass targeted "translation" libraries that transfer existing libraries for the LIT. We then show the quantitative accuracy of targeted PRMs with a Q-LIT using matched-matrix calibration curves collected with 1, 10, and 100 ng total protein to model low abundant immune cell populations. To

facilitate this, we developed an open-source software tool that directly schedules and optimizes PRM assays from DDA and DIA libraries. Finally, we show quantitative consistency measuring low-level biological targets in cytokine-stimulated CD4$^+$ and CD8$^+$ T cells with as little as 1 ng on column. These results suggest that Q-LITs can perform as inexpensive stand-alone instruments for quantitative proteomics, capable of a wide range of measurements without needing high-resolution mass spectrometry.

## Results

Linear ion traps (LITs) are robust, sensitive, and fast mass analyzers, yet these instruments have limited mass resolution. Previously, our lab demonstrated that LITs could be used effectively as stand-alone mass analyzers to measure low-input samples using an Orbitrap Eclipse™ Tribrid mass spectrometer[22]. In that work, we detected approximately 400 proteins from single cells using data-independent acquisition coupled with chromatogram libraries to help make detections[23]. While our Eclipse instrument configuration ignored the high-resolution Orbitrap mass analyzer, we performed those experiments in the context of a high-end Tribrid instrument. Furthermore, the 400 proteins we measured were the easiest to observe but not necessarily the most biologically useful to monitor. While reduced representation approaches[24,25] that quantify a limited panel of easily observed proteins can help infer biological states, significant hurdles must be overcome to predict the expression patterns of unmeasured proteins. As such, directly measuring panels of proteins of interest in low-input samples using targeted proteomics may be preferable to global proteomics.

In this work, we sought to answer three remaining questions. First, by eliminating the Orbitrap, could an affordable Q-LIT mass spectrometer perform at a high level of analytical rigor as a stand-alone instrument for both library generation and targeted proteomics measurement? Second, can a Q-LIT mass spectrometer quantify peptides at and below the level of single cells? Third, can quantitative experiments measure low-level, biologically relevant proteins, such as cytokines and transcription factors, at or below 1 ng? To this end, we assessed several parameters of a hybrid Q-LIT design produced by Thermo Scientific. First, we tested proteome-wide library generation; then, we assessed quantitative linearity using targeted PRM experiments with 100, 10, and 1 ng sample inputs. Finally, we tested sensitivity and measurement consistency in a biological context.

The Q-LIT platform is a versatile instrument that can perform global (DIA) and targeted (PRM) proteomics with the same instrument. Dedicated low-resolution triple quadrupole instruments are capable of highly sensitive measurements with wide dynamic ranges. However, they are limited to selected reaction monitoring (SRM) for specific precursor/fragment ion transitions. The Sciex QTRAP platform is a hybrid triple quadrupole that can scan as an ion trap in the last quadrupole, making it also capable of PRM and DDA. While similar in geometry to the Stellar, the QTRAP is not configurable to perform DIA, in part because of its slower speed. In contrast, Stellar can scan approximately 10× faster than the QTRAP 6500 + , making it an interesting candidate for a low-resolution instrument for developing a stand-alone workflow for transitioning global results to targeted assays. Other high-resolution instruments can also perform both global and targeted scans; therefore, we wanted to compare the Q-LIT to existing instrumentation using the Q-Orbitrap (Exploris 480) as a benchmark. Performance characteristics of other related instruments from a wide variety of vendors are tabulated in Supplementary Table 1, which was modified from a recent literature review by Peters-Clarke et al.[26]. We have provided additional information on operating the Q-LIT in Supplementary Note 1.

### Assessing the performance of a Q-LIT mass spectrometer

The hybrid Q-LIT mass spectrometer has improved ion transmission features capable of performing rapid scans up to 200 kDa/second

(Fig. 1a). The instrument shares many of the same design components as existing Orbitrap-based instruments[27–29]. Ions are passed through a mass filter quadrupole ($Q_1$), then concentrated within the ion routing multipole ($Q_2$) before mass analysis in the LIT. The $Q_1$ mass filter upstream of the LIT is designed to increase ion transmission using an optimized rod shape with hyperbolic surfaces that allow for isolation windows as small as 0.2 to 2 $m/z$ FWHM (typically below 1 $m/z$)[30]. Within this work, we generally maintain a minimum of 2 $m/z$ isolation windows for PRM to capture multiple isotopes per precursor simultaneously, thus increasing sensitivity. These configurations produce high scanning speeds by performing ion accumulation in parallel with mass analysis in the low-pressure LIT[31,32].

We first compared the Q-LIT to a Q-Orbitrap using wide-window DIA. We analyzed HeLa peptides with fixed 8 $m/z$ isolation windows (LIT) or 16 $m/z$ staggered isolation windows[33] (Orbitrap, effectively 8 $m/z$ after demultiplexing) at multiple input levels. Similar to previous work[17,22], we observed the LIT outperformed the Orbitrap at low input (Fig. 1b), and we identified over 6000 proteins (filtered to a 1% FDR) from only 10 ng of input material. With the Q-LIT, the point of diminishing returns with DIA peptide and protein detection is 5× higher than we previously observed with the Eclipse[21], underlining the benefits of faster MS2 scanning. At 50 ng or higher input, the Orbitrap essentially equaled or outperformed the LIT, where access to high mass accuracy became more important than increased sensitivity.

We then wanted to assess the dynamic range of targeted proteomics using the LIT. To do so, we collected a PRM assay targeting Pierce Retention Time Calibration (PRTC) peptides spiked into a HeLa proteome at a ratio of 100 fmol to 100 ng and diluted over 5 orders of magnitude. We acquired the PRTC measurements at 67 kDa/second with a maximum ion injection time of 200 ms and quantified them in Skyline after normalizing to a 500 ng injection (Fig. 1c). We found that the instrument gave a linear signal over 3 orders of magnitude for every PRTC peptide.

## A Q-LIT workflow for generating PRM assays using translation libraries

PRM assays are commonly generated from various sources, including public repositories that store targeted proteomics data such as the PeptideAtlas[34], CPTAC[35], or Panorama[36]. Additionally, assays can be built from global data where targets are selected from empirical measurements within the biological matrix of interest. For this work, we wanted to use methods that could be fully acquired on the Q-LIT yet still be capable of detecting low-abundant peptides. One advantage of this approach is that targets are tuned for the instrument from the context of retention time scheduling and optimal transition selection.

A standard limitation to workflows transitioning assays from high-resolution to nominal-mass accuracy instruments (e.g., Q-LIT and triple quadrupoles) is the need to rebuild any given assay for the platform by re-validating transitions and peptides of interest. We implemented a workflow and software tool to take advantage of the ability to generate peptide libraries using off-line fractionated DDA or gas-phase fractionation data-independent acquisition (GPF-DIA), and developed software to build on-the-fly PRM assays for the same instrument. (Fig. 2a). Briefly, the approach used a DIA search engine to identify library peptides from GPF-DIA data collected on the targeted acquisition instrument with similar acquisition parameters (specifically, 2 $m/z$ isolation windows) and the same HPLC gradient used for PRM analysis. The software was implemented to work with input libraries from a previously generated high-resolution DDA-based spectrum library, peptide predictions using Prosit, or direct DIA search engine results (e.g., Chimerys). This analysis translated existing or predicted spectrum libraries for the PRM instrument by finding when a given peptide elutes in the PRM gradient, identifying instrument-specific fragmentation patterns, and determining which transitions remain quantitative when making low-resolution measurements. Additionally, the software was designed to use retention times from a recently acquired DIA injection to extrapolate a schedule reflecting the current LC column conditions. This collection of PRM-validated peptides, referred to as a

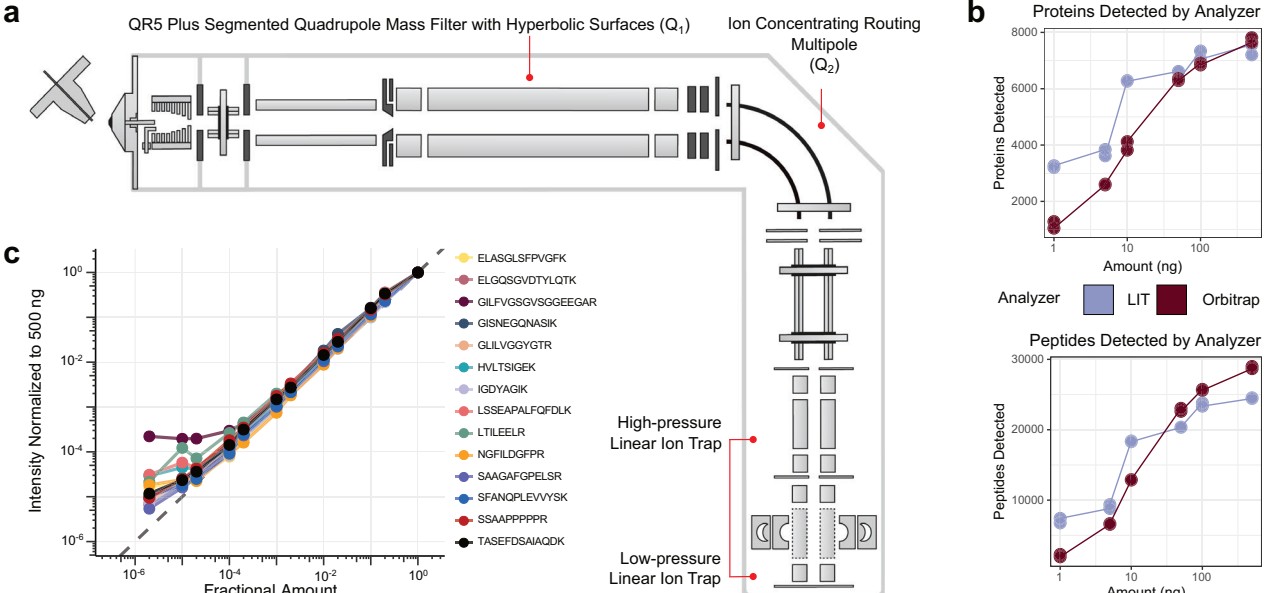

**Fig. 1 | Schematic and performance metrics of the Q-LIT instrument. a** The instrument schematic of the Q-LIT MS. Ions enter the first QR5 Plus Segmented Quadrupole Mass Filter with Hyperbolic surface ($Q_1$) before entering into the Ion Concentrating Routing Multipole (IRM). The IRM behaves as the collision and storage cell. Ions are then moved to the high-pressure cell of the dual-pressure LIT and eventually to the low-pressure cell for mass analysis. **b** The number of HeLa proteins and peptides detected from 1 to 500 ng inputs analyzed with an Orbitrap (Exploris 480) and LIT (Stellar). Each input level was collected in duplicate. **c** The intensities of PRTC peptides in a HeLa background diluted in water over >5 orders of magnitude. Intensities were normalized to 500 ng and scaled to 1. The gray dashed line represents a 1 to 1 fitting between the amount analyzed and the intensity acquired.

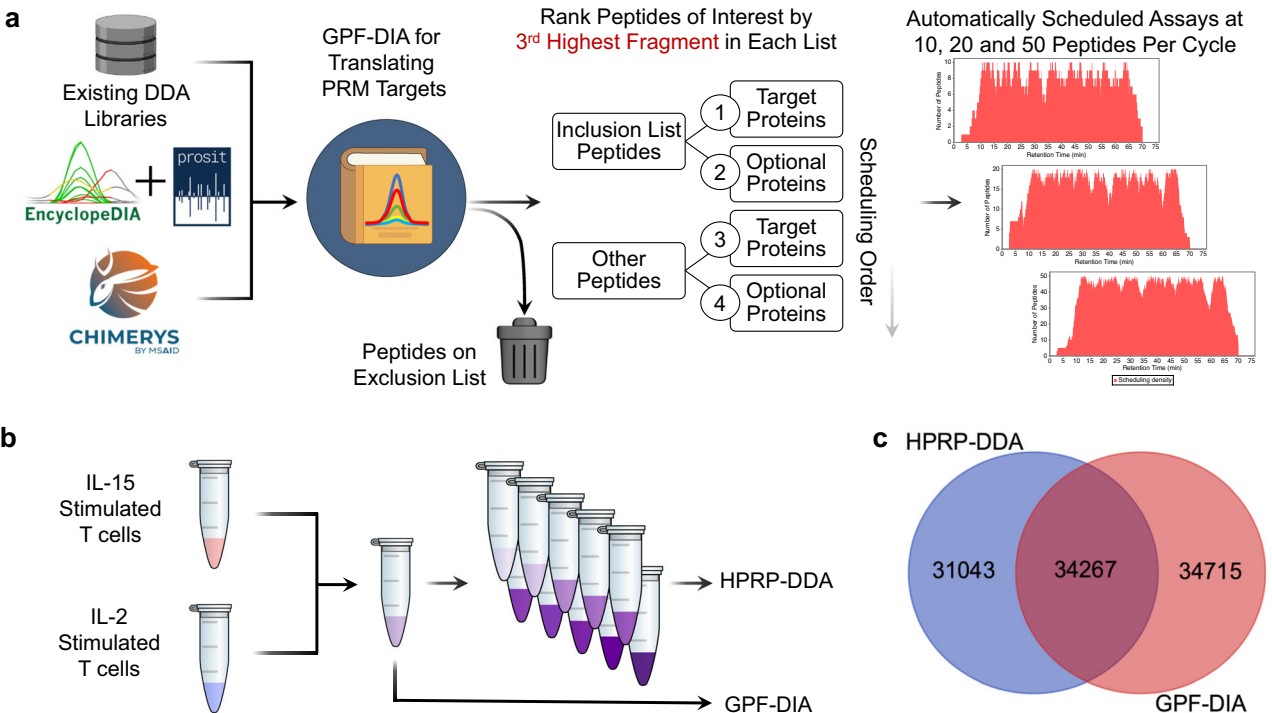

**Fig. 2 | Schematic for generating DDA and DIA-based libraries and detection results. a** A schematic for using GPF-DIA to build targeted PRM assays. A variety of library sources are first mined for library entries, then either re-searched using EncyclopeDIA or inferred using direct DIA search engines, constructing a "translation" library of potential PRM targets and their chemical characteristics. Peptides marked for exclusion are removed from the library, while the remaining peptides are sorted according to their experimental relevancy (using the 3rd highest fragment). For this work, we had the software build PRM assays designed for 1, 10, and 100 ng input levels using 10, 20, and 50 peptides per cycle, respectively. **b** Libraries were generated using either the translation library approach or a more standard DDA method coupled with offline high-pH reverse phase (HPRP) fractionation. **c** The overlap of potential PRM candidates in the spectral library using HPRP-DDA and the translation library filtered to a 1% peptide-level FDR.

"translation library," serves as a database of potential peptides to select for targeted assays. Translation libraries act as DIA chromatogram libraries[23,37] with the purpose of efficiently and quickly translating the chemical characteristics of library entries for target peptides from one instrument/acquisition approach or prediction space to the measurement space of the instrument used for PRMs.

The software, combined with the translation library, is designed to schedule a PRM assay from a list of target accession numbers and other optionally desired accessions from a selected FASTA database. The assay can be modified using both a peptide inclusion and exclusion list. Assays can be adjusted depending on instrument settings, where the maximum assay density and a retention time scheduling window width must be specified. Peptides are ordered based on the third largest fragment ion per peptide, following common SRM/PRM conventions requiring at least three transitions[11]. The algorithm chooses peptides using a greedy approach, where the most abundant peptides are scheduled first based on target preferences indicated through target inclusion and exclusion lists. After the algorithm chooses a specified number of peptides for a given protein (typically 3–5), no additional peptides from that protein are considered. Additionally, peptides cannot be added to a retention time region if any time point in that region has already reached the maximum assay density. Once the algorithm iterates through all possible peptides, the software tool produces a scheduling report and a target inclusion list for the Thermo method editor. While the tool focuses on simplifying scheduling for Thermo instruments, it is analyzer and vendor-agnostic, supporting scheduling for both Orbitrap and ToF instruments. This software workflow has an accessible graphical user interface built into the EncyclopeDIA code base (see Supplementary Note 2 and Supplementary Data 1 for more details). The entire workflow can be performed in a single workday, from translation library to PRM assay. For this work, a single library was generated in 7.5 h, including 6× h-long gradients followed by 15 min for sample loading and column equilibration, where the translation library was processed in parallel with the acquisition.

To generate a low-input PRM assay on the Q-LIT, we tested two standard methods of building libraries: a translation library using GPF-DIA and a spectral library using fractionated DDA. We collected libraries from a pool of IL-2 and IL-15-stimulated T cell proteomes. To build the translation library, 6x gas-phase fractions were used with 2 $m/z$ wide isolation windows across mass ranges of 100 $m/z$ per injection. Since the background proteome matrix is not drastically chemically altered or diluted, this approach produces retention times that closely match the quantitative PRM experiments. In contrast, we offline fractionated the DDA library samples using high-pH reverse phase separations to yield a total of 10 fractions, which were analyzed in separate injections (Fig. 2b). Consequently, each fraction has a simplified matrix background, which may not reflect retention times as consistently in unfractionated quantitative samples. The DDA and DIA methods produced similarly sized libraries, but surprisingly distinct populations of peptides (Fig. 2c), presumably due to the difference in matrix complexity that resulted from the two fractionation methods used to generate each library.

In addition to producing slightly more peptide detections, peptide-centric extraction[38] of DIA datasets is more akin to fragment-level quantification using targeted methods than DDA measurements[39]. As such, peptides detected using GPF-DIA are more likely to produce robust, targeted assays since the mode of discovery uses similar methodologies to the final quantitative measurements. However, some sample types, such as enriched phosphopeptides, may be better suited to library generation with DDA since the semi-

stochastic sampling of precursor ions for fragmentation allows for a greater number of unique positional isomers to be detected when combining technical replicates[40]. For this work, we proceeded with the translation library for assay development, but the scheduling software produced for this work functions with either a DDA or DIA-derived library source.

For DIA injections, search results from EncyclopeDIA and CHIMERYS were combined for downstream work. CHIMERYS is a spectrum-centric search engine that builds on INFERYS to provide spectra and retention-time predictions for peptides in a given FASTA database[41]. In comparison, we searched a Prosit-predicted spectral library[42,43] with the peptide-centric search engine EncyclopeDIA, adapted for analyzing ion trap data. Consequently, EncyclopeDIA was limited to searching +2 and +3 peptides to maintain a reasonable search space, while CHIMERYS was configured to consider modifications and higher charge states. More peptides were detected from CHIMERYS compared to EncyclopeDIA in each gas-phase fraction (Supplementary Fig. 1a), yet considering this superset of detections increased the total number of potential targets (Supplementary Fig. 1b). Both search engines produced an equal number of viable peptide targets that could be used in downstream PRM experiments (Supplementary Fig. 1c). In all cases, the retention times from CHIMERYS-detected peptides were re-peak picked using EncyclopeDIA to identify candidate target transitions for PRM measurement in a combined DIA library.

## Assessing Q-LIT PRM quantitative accuracy at low input

With low-input global proteomics, we preferentially measure only the most abundant proteins. We stress-tested the quantitative accuracy of the Q-LIT system using PRMs by measuring biologically relevant proteins that tend to occur at a range of abundances in the proteome. To accomplish this, we first functionally annotated candidate peptides in the combined DIA library using the PANTHER database[44]. We selected target proteins based on GO-terms and Reactome pathways for T cell differentiation, immune biology, T cell activation, cytokines, and transcription factors, focusing on selecting proteins associated with the dynamics of memory or effector T cells. Using the PRM scheduling algorithm, we constructed three assays using the same bank of proteins, where each assay was suited to a different input level: up to 50 peptides/cycle for 100 ng of material, 20 peptides/cycle for 10 ng of material, and 10 peptides/cycle for 1 ng of material. Ultimately, the 100 ng assay quantified 481 peptides, the 10 ng assay quantified 151, and the 1 ng assay quantified 61. To maintain a 2-second cycle time using 1 ng of material, the maximum ion injection time (maxIIT) was set to 200 ms. Similarly, at 10 ng of material, the maxIIT was set to 95 ms (slightly below 100 ms to accommodate the additional time required to route ions in the mass spectrometer). At 100 ng of material, the ion injection time was set to 50 ms; however, each scan rarely met that length of time.

We performed matrix-matched calibration curves[45] at 100 ng, 10 ng, and 1 ng levels to assess the quantitative accuracy of the Q-LIT over several orders of magnitude. Dilutions in a buffer background are helpful to assess instrument sensitivity, but because background noise decreases at the same rate as target peptides, quantitative linearity will always appear more accurate than in a real background matrix. Matrix-matched calibration curves are more effective at assessing linearity in real-world scenarios since the background signal does not change with dilution. To accomplish this, we had to build a suitable background matrix of similar composition to our target T cell proteome. Our approach used dimethyl labeling to modify the foreground T cell proteome, keeping the same composition while producing different precursor and fragment masses. Dimethyl labeling was first introduced as a multiplexing method where multiple samples would be labeled and mixed prior to mass spectrometry[46]. In our approach, only the background is modified, where free amines are mass-shifted by two

methyl groups (+28 Da). This shifts any labeled precursors (even incomplete reactions with a single methyl group) outside the precursor isolation window used by PRM measurements, ensuring that foreground signals will not be confused with background signals. Additionally, dimethyl labeling is affordable, easy, and quick, as peptides are labeled to 99.9% completion within a 1-h reaction.

From this experiment, we found that reasonable quantitative accuracy (Supplementary Fig. 2) can be achieved with the Q-LIT at low input. At 100 ng, the quantitative accuracy of most peptides acquired with PRM remains consistent for nearly two orders of magnitude (Fig. 3a), where the median lower limit of detection (LoD) was 0.83:100 (ratio of foreground to background) and the median lower limit of quantification (LoQ) was 2.8:100 (Fig. 3b), where only 0.6% of peptides could not be assigned a LoQ. Quantification was slightly worse at the 10 and 1 ng levels, where 4.6% and 20% of peptides could not be assigned a LoQ (Fig. 3c). The expected target ratios are annotated in Fig. 3d. As the analyte signal dropped with decreasing concentration, background interference tended to overwhelm the analyte signal. As a result, quantitative ratios with the Q-LIT tend to regress to 1:1, resulting in higher-than-expected measured ratios. Unsurprisingly, at the 100 ng level, the signal is more easily distinguishable from noise, and the LoD distribution is generally lower than at 10 ng or 1 ng (Fig. 3e and f).

We acquired analogous PRM assays on a Q-Orbitrap Exploris 480 at 1, 10, and 100 ng using the majority of peptides that were quantified in the calibration curve assays for the Q-LIT. Q-Orbitraps have limited trapping capacity, which can limit the dynamic range within a spectrum. In contrast to the Q-LIT, peptides with low background signal simply stop being measured before they fall below the LoQ, resulting in quantitative ratios regressing to 0:1 and lower than expected measured ratios (Supplementary Fig. 3). A consequence of this signal drop-off is that estimated LoQ and LoD values for the Q-Orbitrap can be more difficult to estimate correctly, producing modes around samples where the peptide signal falls below the spectrum dynamic range (Supplementary Fig. 4). Although the distribution of LoDs remains similar between the Q-LIT and Q-Orbitrap, the distribution of LoQs is typically lower for the Q-LIT on a peptide-by-peptide basis (Supplementary Fig. 5). In addition to producing improved LoQs, more peptides could be targeted with the Q-LIT since the instrument has faster scanning speeds. For example, at 100 ng level, 300 peptides were scheduled on the Q-Orbitrap, while 473 peptides could be scheduled on the Q-LIT before the schedule was at maximum capacity.

Single cells typically produce between 0.1 and 0.3 ng of peptides, depending on the cell type. Considering the 1 ng sample, the median measured peptide produced a linear signal in this range (0.198:1). Several peptides showed a linear response below 0.1 ng. For example, the peptide ECESYFK from Granzyme B was found to have a LoQ of 0.043:1, equating to a proteome fraction consisting of 43 pg in a background of 1 ng, and was still measurable above background at the 18 pg level (Fig. 4a and b). Two other Granzyme B peptides, VAA-GIVSYGYK and TQQVIPMVK, had produced even lower LoDs (below 10 pg equivalents). Granzyme B is a serine protease implicated in multiple autoimmune diseases[47]. All told, 61 peptides with estimable LoQs in the 1 ng assay corresponded to 30 quantified proteins with a median of 11-14 points across the peak base using the Q-Orbitrap and Q-LIT (Supplementary Fig. 6).

## Validated cell populations for quantitative testing

In addition to showing quantitative accuracy in a controlled matched matrix, we wanted to validate measurement precision in low-input biological experiments. The interleukins (IL) family of proteins is a class of cytokines expressed by many cells, including immune cells, which bind to specific receptors that elicit pro- and anti-inflammatory roles[48]. Certain cytokines, such as IL-2 and IL-15, bind to receptors on the surface of T cells in specific biological events, such as activation and differentiation. Both of these molecules have been successfully

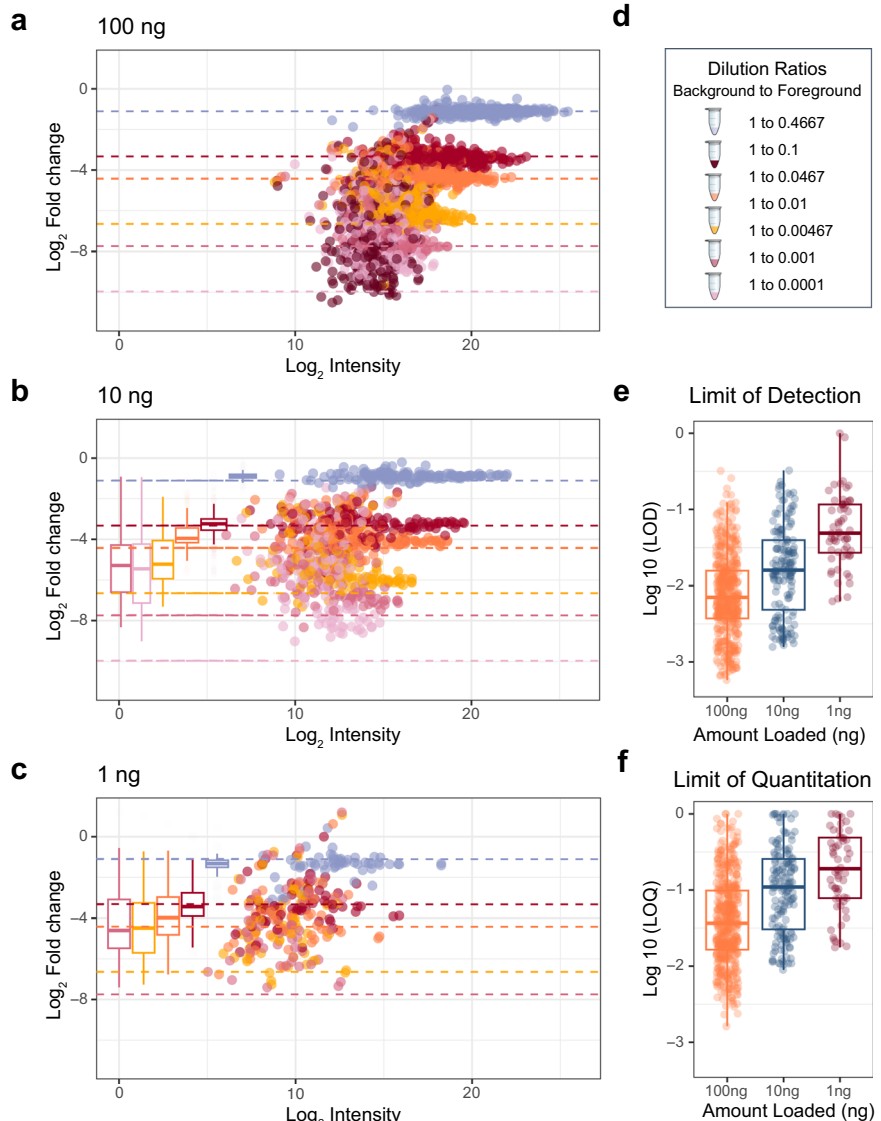

**Fig. 3 | The figures of merit for PRM curves at 1, 10 and 100 ng of material.**
**a**, **b** The quantitative accuracy of matrix-matched curves on an ion trap of pooled IL-2 and IL-15 peptides in a background of dimethyl-labeled peptides. We generated three curves loading 100 ng, 10 ng, and 1 ng of material on-column. Box plots show the spread of measured values where the whiskers indicate 5% and 95% points, and the bold line indicates the median measurement. **d** Each dilution is a different color where colored dashed lines indicate the expected fold change. **e**, **f** The distribution of the Figures of Merit for the 1, 10, and 100 ng injections using PRM on the Q-LIT. All boxplots (**a**–**c**, **f**) are represented as median value. The box maxima extents to the 1st interquartile range (25th percentile), while the minima extends to the 3rd inner quartile range (75th percentile).

used as part of immunotherapies to combat cancer[49–52]. Interestingly, IL-2 and IL-15 are structurally similar in homology and activate T cells through the same receptor subunits (IL-2/IL-15Rβγ)[53,54], mediating largely similar biological effects on T cells[55,56]. However, possibly related to the expression of the private IL-2Rα and IL-15Rα chains, IL-2 induces an effector-like phenotype (with low CD62L expression) while IL-15 induces a memory-like phenotype (with higher CD62L)[57]. We generated activated T cells cultured in IL-2 or IL-15 to replicate an effector-like and memory-like phenotype for CD4+ and CD8+ cells (Supplementary Fig. 7a). We selected this model system to showcase the ability to generate LIT-PRM assays using well-studied biology at inputs below 1 ng. Additionally, flow cytometry was used as an orthogonal technique to validate the cell populations present in IL-2 and IL-15 treated T cells on days 5, 6, and 10, which exhibited an effector-like and memory-like phenotype (Supplementary Fig. 7b and c).

At day 10, flow cytometry identified that each culture was predominantly composed of T cells, with CD8+ T cells being the majority subset in both IL-2 (83.8%) and IL-15 (92.2%) cultures (Fig. 5). Correspondingly, we found that CD4+ T cells composed 14.6% of the cells stimulated with IL-2 and 6.9% of the cells stimulated by IL-15. This was reflected in our targeted proteomics data, as the CD4 protein was the second most downregulated protein in IL-15-stimulated T cells compared to IL-2-stimulated cells (Fig. 6a). We note that this protein was not technically quantified at 1 ng, as the one peptide for CD4 (VVQVVAPETGLWQCLLSEGDKVK) lacked linearity in signal as estimated by the calibration curve. We measured the same peptide at the 10 ng level, where we calculated the LoD to be 0.96:10 (ratio of foreground to background) with an LoQ of 8.3:10 (Supplementary Fig. 8). This indicated that at the 1 ng level, CD4 should be above the LoD but below the LoQ, and our results match these calculations.

IL-2 stimulation is known to push activated T cells into an effector-like population, which is reflected by the paired flow cytometry data on day 10 of treatment with recombinant human IL-2. Granzyme B, from which we estimated the most responsive peptide (TQQVIPMVK) was

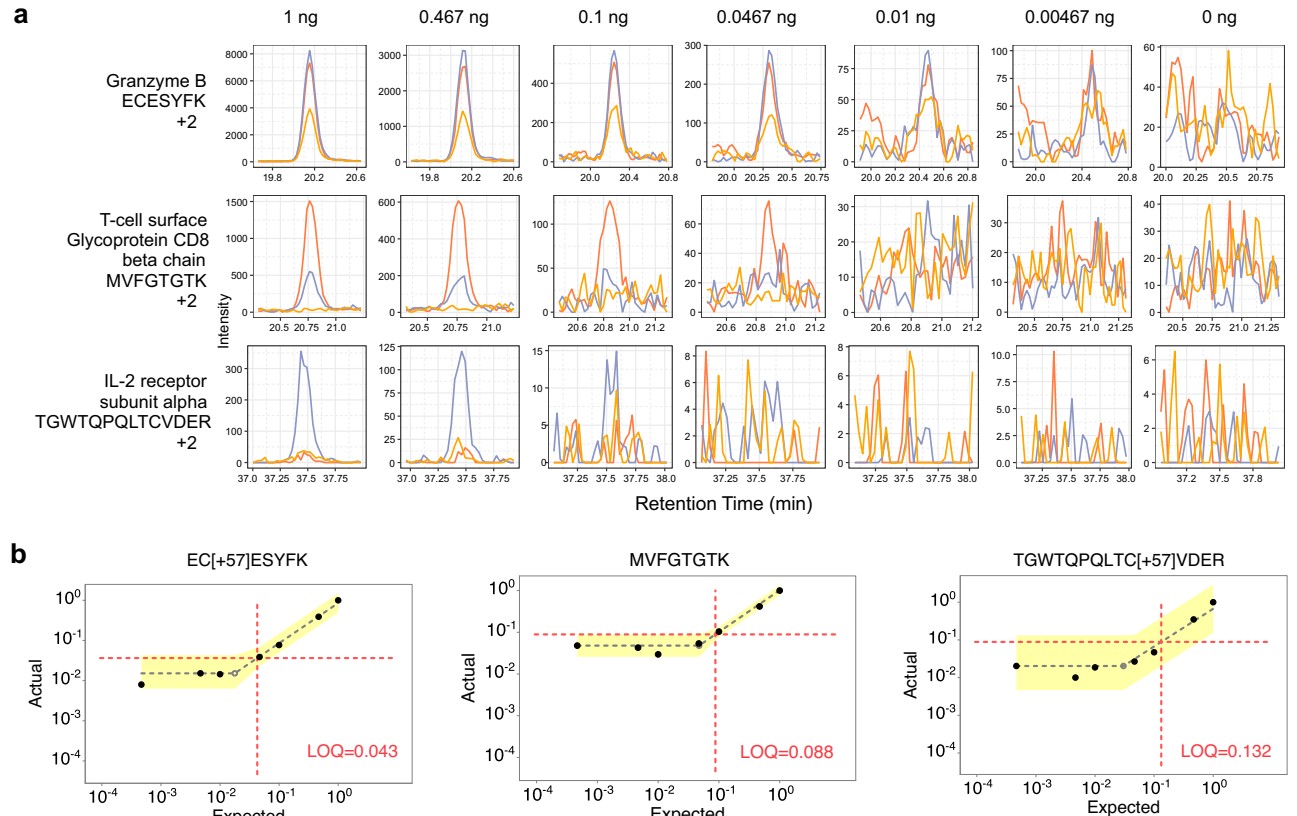

**Fig. 4 | Three representative peptides that were quantifiable below 1 ng. a** Each row displays a peptide chromatogram at each dilution within the 1 ng curve. Each peptide contains three representative transitions. The first peptide from Granzyme B had the best estimated LoQ at 0.043:1, while the third peptide from IL-2 receptor subunit alpha had an estimated LoQ at 0.132:1 at 1 ng. **b** LoQ and LoD were estimated on a peptide-by-peptide basis using EncyclopeDIA's curve fitting algorithm. First, the algorithm determines a maximum line through the noise of the calibration curve and then fits the linear dynamic range. The intersection of both lines is the LoD (shown with a gray shaded, empty circle), while the LoQ (shown at the intersection of the dotted red lines) is three standard deviations of the noise above the LoD. The error associated with the lines fitted through the noise and linear, dynamic range are shown in yellow, and represent 3 standard deviations above and below the median signal for each line.

quantitative to 0.025:1 (ratio of foreground to background), is an effector molecule secreted by cytotoxic CD8$^+$ T cells. We found peptides associated with this protein were 1.55x lower in IL-15 than IL-2 stimulated cells using targeted proteomics (Fig. 6a), which matches flow cytometry data indicating that the number of effector CD8$^+$ T cells ($T_{EFF}$) is lower when stimulated with IL-15 than IL-2. While both IL-2 and IL-15 resulted in T cell activation, IL-15 stimulation differentiated memory-like T cells, as demonstrated in the flow cytometry data. The CD44 receptor antigen is a cell surface receptor that helps cells facilitate cell-cell interaction and response to the tissue microenvironment. Interestingly, we found that the expression of CD44 is slightly higher in IL-2 compared to IL-15, indicating that IL-2 stimulated cells had a higher population of activated cells, with a 1.6x median fold change in abundance. T cells that express CD62L have an increased population of memory T cells after IL-15 stimulation[54]. Flow cytometry data indicated that we had a higher population of CD62L$^+$ cells in the IL-15 stimulated condition than T cells stimulated with IL-2 (Supplementary Fig. 7), specifying a higher population of memory-like T cells. Compared to the IL-15 stimulated cells, IL-2 stimulated T cells expressed IL-2Rβ/IL-15Rβ at a higher ratio (Fig. 6a), which is associated with a memory phenotype. In general, we observe high analytical precision using PRM with a Q-LIT platform, even in 1 ng assays. The majority of peptides were measured with less than a 20% coefficient of variation between 3 technical replicates (Fig. 6b), indicating quantitative rigor within the workflow.

Ultimately, we detected 100% of the proteins monitored with flow cytometry using global proteomics during library generation. While some of these proteins were hard to observe at low input (1 ng), we were able to quantify 75% above an estimated LoQ with targeted proteomics. This overlap indicates complementary benefits of using flow cytometry in tandem with targeted proteomics to capture the immune cell state fully. While single-cell proteomics using mass spectrometry continues to develop, flow cytometry is the best method for measuring a small number of proteins (6-12) on thousands of individual cells within a single day. On the other hand, targeted mass spectrometry on 1-10 T cells (equivalent to around 0.1 and 1 ng) can monitor tens to hundreds of proteins, including cytokines and transcription factors, which cannot be easily monitored using flow cytometry.

## Discussion

Here, we demonstrate a complete workflow for acquiring global libraries and rapidly building de novo PRM assays using only a Q-LIT mass spectrometer. While the Q-LIT is capable of DDA, we found that equivalently large libraries could be quickly generated using GPF-DIA. We also found that PRM assays using the Q-LIT were linearly quantitative even at low input, enabling us to accurately monitor difficult-to-measure cytokines, transcription factors, and immune proteins. From a broader perspective, high-resolution mass spectrometry is expensive in terms of instrument costs, high vacuum, and power requirements, often requiring greater technical experience to operate successfully. In contrast, Q-LIT mass analyzers are easier to maintain and more cost-effective to operate in part because they have less stringent vacuum requirements compared to Orbitrap-based mass spectrometers, making them an appealing option for low-input proteomics. These

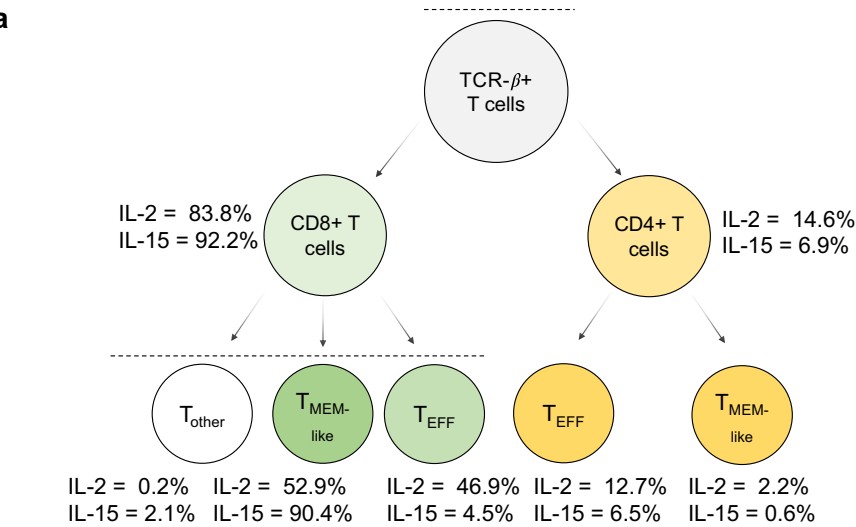

**Fig. 5 | A summary of the cell populations in IL-2 and IL-15 stimulated T cells determined by the flow cytometry panel described in Supplementary Table 2. a** The gating procedure was used to determine the relative percentage of each cell type in the IL-2 and IL-15 samples (more details in Supplementary Fig. 7). **b** The estimated cell populations based on back calculations of the gating results.

factors are especially important in single-cell proteomics, where each biological sample needs thousands of injections. We believe these instruments offer a high value-to-expense ratio, potentially providing more affordable mass spectrometry instrumentation in a broader array of laboratory settings. This affordability is particularly impactful in immuno-oncology, where proteome-based analysis of immune cell populations can uncover crucial biomarkers to guide clinical decisions.

## Methods

All research complies with the relevant ethical regulations. Work with animals was performed according to the guidelines of the Institutional Animal Care and Use Committee at the Ohio State University.

### T cell cultures

Mice were kept in housing conditions consistent with the ethical guidelines recommended by the IAUAC at the Ohio State University. The humidity was kept between 20-30%, the temperature was maintained between 20-25 °C and the mice are subjected to a 12-h dark/light cycle. Splenocytes from two female C57BL/6 mice were stimulated with plate-bound anti-CD3 mAb (145-2C11 clone) on day 0 in complete media, as described previously[58]. On day 2, cells were washed and re-plated with human (h) IL-2 or hIL-15 at 200 ng/mL. Cells were washed and split on days 4 and 6. Flow cytometry was performed on days 5, 6, and 10, and cells were washed three times with PBS, centrifuged at 500 RCF for 5 min, pelleted, and stored at -80 °C on days 6 and 10 for mass spectrometry. A third condition was also maintained without

stimulation as a control for flow cytometry. For each condition, one representative well was used for flow cytometry analysis. All technical replicates grown in cell culture were pooled together for mass spectrometry analysis.

### Flow cytometry

Flow cytometry was performed as previously described[58]. Briefly, cells collected on days 5, 6, and 10 were stained with live/dead fixable blue dead-cell stain (Invitrogen #L23105), and antibodies for B220, CD4, CD8, CD25, CD44, CD62L, CD69, and TCRb (see antibody details in Supplementary Table 2). Stained cells were acquired with a Cytek Biosciences Aurora™ 5-laser flow cytometer and analyzed using BD Biosciences FlowJo™ software.

### Proteomics sample preparation

Frozen cell pellets were lysed in a 5% SDS buffer containing 50 mM TEAB, 1x HALT, and 2 mM MgCl$_2$. DNA was sheared with a Bioruptor® Pico by sonicating at 14 °C for 30 seconds, followed by 30 seconds of rest, a total of 10 times. Sheared cells were then spun down at 13,000 RCF for 10 min, and the protein supernatant was retained. Protein quantities were estimated using a Pierce™ bicinchoninic acid (BCA) Protein Assay Kit. Proteins were reduced with 40 mM dithiothreitol (DTT), alkylated with 40 mM iodoacetamide, and quenched with 20 mM DTT. Acidification was done with 2.5% phosphoric acid, and protein was loaded onto suspension trap (s-trap) micros (Protifi LLC). Digestion was performed with trypsin at a 1:20 ratio of enzyme to

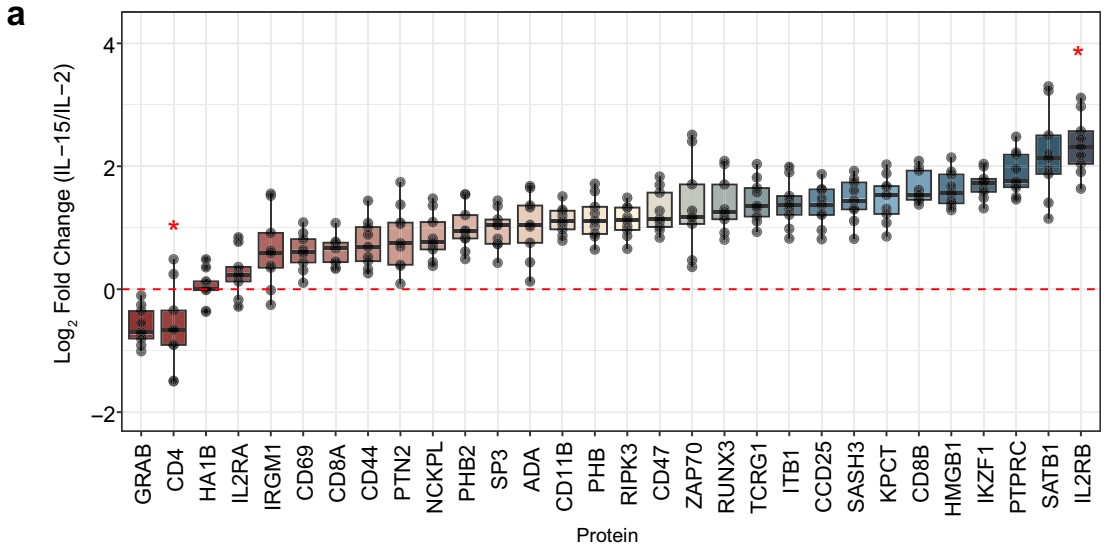

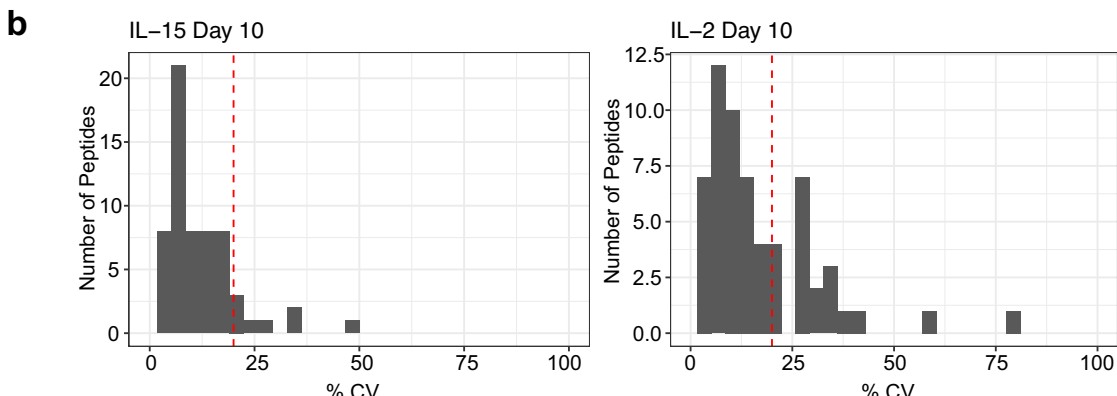

**Fig. 6 | Quantifying immune cell biological replicates at 1 ng. a** Quantitative ratios for the panel proteins assayed in the 10 peptide/cycle PRM. The assay was collected in technical triplicate injections of Day 10 IL-2 and IL-15-stimulated T cell proteomes. The selected panel of proteins is associated with T cell activation, differentiation, or cytokine signaling. No LoQ was determined for CD4 with the 1 ng calibration curve, indicated by a red 5-point star. In the IL-2 stimulated sample, IL2RB was measured below the LoQ determined by the 1 ng calibration curve, indicated by a 6-point star. The 9 data points of each protein were extracted from 3 technical replicate PRM injections for the IL-2 and IL-15 stimulated proteomes by calculating the log2 fold change in all possible combinations using technical replicates. The boxplot centers are represented as median values. For each box, the maxima extents to the 1st interquartile range (25th percentile) and the minima extends to the 3rd inner quartile range (75th percentile). **b** Coefficient of technical variation (% CV) plots for all peptides quantified in the 1 ng assay. The red dotted line indicates 20% CV on each plot.

protein at 47 °C for 2 h, then eluted. Peptides were dried down and stored at -80 °C.

According to the kit protocol, an aliquot of dried peptides was separated according to basicity using a Pierce High pH Reverse-Phase Fractionation Kit. Briefly, 50 µg of peptides were resuspended in 0.1% trifluoroacetic acid in HPLC-grade water. The separation mini-columns from the kit were centrifuged at 5000 RCF for 2 min to remove any liquid and pack the resin. The mini-columns were then equilibrated with 100% acetonitrile and washed 3 times with water. Resuspended peptides were loaded, and the flow through was collected as the first fraction. The mini-columns were washed with water, and the eluent was collected as the second fraction. The elution buffers specified from the kit were then used to produce the following 8 fractions. Fractionated peptides were then dried down and stored at -80 °C until mass spectrometry-based analysis for DDA-based library generation.

A separate aliquot of the eluted peptides were dimethyl labeled using an in-solution amine-labeling reaction published by Boresema et al.[46]. Digested peptides were resuspended in 100 mM TEAB (pH = 8.5). Formaldehyde (4%) was added to the resuspended peptides and mixed. Sodium cyanoborohydride (0.6 M) was then added to catalyze the dimethyl labeling reaction for 90 min at 22 °C while mixing vigorously. The reaction was quenched with 1% ammonia and 5% formic acid. All peptides were resuspended in 2% acetonitrile with 0.1% formic acid. Calibration curves were generated by mixing labeled and unlabeled peptides at different concentrations. In these mixtures, unlabeled peptides were diluted in a dimethyl-labeled background over 4 orders of magnitude (Supplementary Table 3) and aliquoted at different concentrations prior to mass spectrometry analysis.

### LC-MS settings

Data was acquired on a Thermo Scientific™ Stellar™ MS coupled to a Vanquish™ Neo UHPLC system. Solvent A consisted of 100% water with 0.1% formic acid, and solvent B contained 80% acetonitrile with 0.1% formic acid. An Easy-Spray™ source was used for ionization at 2000 V, and the ion transfer tube was set to 275 °C. Peptides were separated on a 25 cm C18 analytical Easy-Spray column, packed with 2 µm beads along a 60-minute linear gradient as follows: from 0-4 min, 2% B, 4-8 min increased to 8% B, 8 to 58 min increased with 28% B, 58 to 65 min increased to 44% B, followed by a 10-minute wash at 100% B. The flow for the entire gradient was set to 250 nL/min. The instrument was

configured to expect chromatographic peaks of approximately 15 seconds, fragment peptides with a default charge state of +2, and a collision cell gas pressure of 8 mTorr.

## Q-LIT DDA acquisition

For DDA experiments, the RF lens was set to 30%. Precursor spectra ranged from 350-1250 $m/z$ at a scan rate of 67 kDa/second. The automatic gain control (AGC) target was set to "Standard" with an absolute AGC target of 3e4. The maximum ion injection time (maxIIT) was set to 100 ms, and spectra were collected using centroiding in positive mode. MS2 scans were collected only on peptides with a charge state greater than +1, excluding undetermined charge states. An intensity threshold of 5E2 was used to trigger an MS2 scan and an HCD NCE of 30%. Following the MS2 measurement, the peptide m/zs were placed on a dynamic exclusion list for fragmentation for 2 seconds using a precursor mass tolerance of +/- 0.5 $m/z$. Twenty DDA scans were taken in each cycle with a 1.6 $m/z$ isolation window around the precursor of interest. Fragment ions were scanned at 125 kDa/second scan rate from 200-1500 $m/z$ using an AGC target of 1E4 and a maxIIT of 50 ms.

## Q-LIT DIA and PRM acquisition

For both DIA and PRM experiments, the precursor range was set to 350-1250 $m/z$ and measured at a rate of 67 kDa/second. The AGC target was set to "Standard," which is equivalent to 1e4, and the maxIIT was set to 100 ms. The loop control was set to "all." Peptides were fragmented with HCD with NCE set to 30%, and fragments were scanned at 67 kDa/second over a 200–1500 $m/z$ range. Precursor isolation windows for DIA were consistently 8 $m/z$ wide, where margins were set to forbidden zone locations.

For GPF-DIA, six gas phase fractions were collected over 400–500, 500–600, 600–700, 700–800, 800–900 and 900–1000 $m/z$. The majority of settings were the same as a wide-window DIA scan, except for 2 $m/z$-wide isolation windows over the adjusted precursor $m/z$ range for each method.

All settings for PRM scans were the same as for wide-window DIA, except for the isolation window width and maxIIT. For PRM assays at 10, 20, and 50 peptides per cycle, the maxIIT was set to 200, 95, and 50 ms, respectively. Precursor isolation windows were set to 2 $m/z$, where MS/MS were collected over 200–1500 $m/z$. All PRM assays were schedule using EncyclopeDIA-v.4.7.11. Additional information about scheduling can be found in Supplementary Note 2 and Supplementary Data 1.

## Q-Orbitrap DIA and PRM acquisition

All experiments performed on the Q-Orbitrap were performed on a Thermo Scientific Orbitrap Exploris 480 mass spectrometer coupled to a Thermo Scientific Easy nLC-1200. The LC buffers and gradient matched those used on the Q-LIT. For DIA injections and generating a translation library, the isolation windows and ion injection times used were similar to those used on the Q-LIT except for resolution, AGC target, and ion injection time. On the Exploris for DIA scans, the MS1 scan had an AGC target set to 3E6 ions, or 300%, while the ion injection time was set to "auto". The MS1 resolution was set to 60k, while the MS2 resolution was 15k. The MS2 scan used a "DIA" scan, had the AGC set to 1000%, which is equivalent to 1E6 ions, and ion injection time was set to "auto" for all DIA injections.

For PRM injections on the Orbitrap, all settings were the same as those used in the Q-LIT; however, the MS2 resolutions were altered to account for differing amounts of material loaded. At all levels, the MS1 settings matched what was used for DIA injections on the Orbitrap, where the MS1 resolution was maintained at 60k. For the 100 ng PRM assay, the MS2 resolution was set to 60k. At 10 ng of material, a 30k resolution was used, and at 1 ng of material, a 15k resolution was used to get the necessary scan speed to a similar number of targets as used on the Q-LIT.

## Data analysis

Global data was first converted to the universal mzML format using peak picking. DIA data was analyzed with EncyclopeDIA v.4.7.11 using the "IonTrap/IonTrap" or "Orbitrap/Orbitrap" mode. Mass tolerance was set to 0.4 Da, where a minimum of 3, but a maximum of 5 ions were used for quantification. The translation library was generated by searching 6 gas phase fractions against a Prosit[42] predicted library. The Prosit library contained spectrum predictions of all +2 and +3 ions from a mouse FASTA from UniProt, which was accessed on October 22, 2019. The predicted library allowed for up to 1 missed cleavage, with a default charge state of 3, and default NCE of 33 over 396.4–1002.7 $m/z$. Wide-window (8 $m/z$ on the Q-LIT or 16 $m/z$ staggered on the Q-Orbitrap) injections were searched against the translation library using the same search settings. Global DDA data was searched in Scribe using the "IonTrap/IonTrap" instrument mode for b and y tryptic peptides, with a library mass tolerance of 0.4 Da. The 10 high-pH fractionated injections were searched against the same Prosit predicted library used to generate the DIA library. Global DIA data collected with the Pierce HeLa Standard Proteome and containing PRTC peptides spiked in were searched against the pan-human library, with a background FASTA accessed on April 25th, 2019.

Global DIA data was searched using CHIMERYS[59] intelligent search algorithm (MSAID GmbH) in Thermo Scientific™ Proteome Discoverer™ 3.1, using analogous settings for the EncyclopeDIA search. A predicted spectrum library was generated from the mouse fasta database by INFERYS™ deep learning framework (MSAID GmbH) for all tryptic +2, +3, and +4 peptides between 7–30 amino acids in length. For processing, spectrum files were selected using the ion trap MS setting, with a signal-to-noise peak threshold of 1.5. The top 24 peaks were selected in each window with a fragment mass tolerance set to 0.4 Da. Fixed carbamidomethyl modifications and a maximum of 2 oxidized methionines were allowed per peptide. The retention times from CHIMERYS were extracted for all detections and combined with EncyclopeDIA's detections. For peptides detected by both software tools, the EncyclopeDIA retention times were preferred. The combined detections and retention times were used to select peaks within EncyclopeDIA and run against a 1% FDR to obtain a combined search engine library. The fractionated injections were also searched in Proteome Discoverer, using a mass tolerance of 0.4 Da for all +2, +3, and +4 peptides, and the same settings used for the CHIMERYS search, Skyline[60,61] version 23.1.0.455 was used for targeted analysis.

For analyzing the calibration curves and other PRM injections of IL-2 and IL-15 replicates, the GPF-DIA library was first imported to serve as a reference point for integrating low-input PRMs. With the imported DIA results, transition settings were subsequently altered, and the PRM samples were imported. For both imports, the same peptide settings were set to Trypsin [KR\P], with a maximum of 2 missed cleavages, and the mouse fasta used to generate the Prosit library was used to generate a background proteome. Retention time window predictions were set to 5 min; however, measured retention times were used when present. Peptides between 7 and 40 amino acids in length were used, and "auto-select all matching peptides" was left checked. Only carbamidomethylation modifications were considered for cysteine. For transition settings, peptides of +2, +3, and +4 precursor charges, along with +1 and +2 fragment ion charges, were considered for b and y ion types. For product ion selection, the third ion to the second to last ion was considered in Skyline. DIA precursor windows were used for exclusion when importing the GPF-DIA library. The ion match tolerance for the library was set to 0.4 Da, and 6-9 product ions were used from filtered product ions. For the instrument parameters settings, a 200–1500 $m/z$ range was considered, with a method match tolerance of 0.4 Da, and "dynamic min product $m/z$" and "triggered chromatogram acquisition" were checked. Under the full-scan parameters, "DIA" was used when importing the GPF-DIA library file as a reference point for integrating calibration curves. The gas-phase fractionated isolation windowing

scheme was imported from the files for a QIT mass analyzer, with a resolution of 0.4 Da and retention time filtering within 5 min of MS/MS IDs. For importing PRM injections, the "PRM" acquisition method was used under the full-scan settings rather than the "DIA" method. Finally, lower limits of detection (LoD) and quantification (LoQ) were estimated using EncyclopeDIA, and calculated quantities of IL-2 and IL-15 replicates (n = 3) were determined using calibration curves.

## Reporting summary

Further information on research design is available in the Nature Portfolio Reporting Summary linked to this article.

## Data availability

The proteomics raw data and fully processed Skyline documents are available in the Panorama database and are accessible with the following url: https://panoramaweb.org/StellarIonTrapForLowInput.url. Additionally, all proteomics raw data is publicly available on the MASSIVE repository under the accession number MSV000094904. The ProteomeXChange number can be found under the accession number PXD052847. Source data are provided with this paper.

## Code availability

Open-source software developed for this project is publicly available as part of the EncyclopeDIA project at https://bitbucket.org/searleb/encyclopedia/downlaods under "encyclopedia-4.7.11-executable.jar."

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

## Acknowledgements

This research was supported by NIH R35GM150723 and R21CA267394 to BCS. This research was also supported by The Ohio State University Comprehensive Cancer Center (OSUCCC) and the National Institutes of Health (NIH) under grant P30CA016058. This research was made possible through resources, expertise, and support provided by the Pelotonia Institute for Immuno-Oncology (PIIO), which is funded by the Pelotonia community and the OSUCCC. We thank the PIIO and the Immune Monitoring and Discovery Platform for flow cytometry access.

## Author contributions

A.E.S. performed the mass spectrometry sample preparation, data acquisition, and analysis with the help of B.C.S. B.C.S. and A.E.S. designed the concepts within this paper with the help of L.R.H., C.C.J., P.M.R., and M.P.R. R.N.T. performed the animal work, flow cytometry, and cell culture of stimulated T cells. M.P.R. and R.N.T. analyzed the flow cytometry data. Z.L. and N.J.S. provided valuable insights into data interpretation. A.E.S. and B.C.S. were primarily responsible for writing the text, but all authors contributed to editing the manuscript.

## Competing interests

B.C.S. is a founder and shareholder in Proteome Software, which operates in the field of proteomics. The Searle Lab at Ohio State University has a sponsored research agreement with Thermo Fisher Scientific, the instrumentation manufacturer used in this research. However, analytical methods were designed and performed independently of Thermo Fisher Scientific. L.R.H., C.C.J., and P.M.R. are Thermo Fisher Scientific employees, the instrumentation manufacturer used in this research. The remaining authors declare no competing interests.
