## [Transparent Peer Review file · Nature Communications]

Rapid assay development for low input targeted proteomics using a versatile linear ion trap

Corresponding Author: Dr Brian Searle

Version 0:

Reviewer comments:

Reviewer #1

(Remarks to the Author)

In this manuscript, Shannon et al. present a rapid method for developing targeted proteomics assays for low-input materials using a quadrupole-ion-trap mass spectrometer. The authors outline a workflow that involves generating data-dependent acquisition (DDA) and gas-phase fractionated data-independent acquisition (DIA) libraries at low resolution. This approach is complemented by the automatic prioritization of detectable peptides associated with the proteins of interest, while maintaining the number of targets per cycle to ensure reliable quantification. The utility of this workflow is effectively demonstrated by successful quantification of low-level proteins in PRM mode from just 1 ng of sample derived from stimulated T-cells.

The research presented in this article is executed to a high standard, which is consistent with the authors' established reputation in the field and their strong command of the analytical techniques employed. While the work showcases valuable techniques, it appears to build upon methods that have been previously described, often by the authors themselves in earlier publications, and applies them within a q-LIT instrument. These techniques include the creation of libraries, algorithms for prioritizing detectable peptides, and the use of matrix-matched calibration curves for assessing quantitative capabilities. While the application of these methods to the q-LIT instrument is certainly noteworthy, a comparison of key figures of merit with other similar systems could further highlight the innovative aspects of the Stellar. For instance, how does the Stellar compare to the Orbitrap Ascend when operated solely with the quadrupole and linear ion trap? Additionally, how does it perform against other low-resolution instruments capable of PRM-like measurements from different vendors, or even against available high-resolution instruments from the same vendor?

Addressing this aspect could significantly enhance the manuscript's impact and innovation aspect. Therefore, while I appreciate the quality of the research, I would suggest further exploration of these comparisons before considering publication in Nature Communications.

(Remarks on code availability)

There is no easily identifiable release on the main page corresponding to the version v3.0.0-SNAPSHOT mentioned in the "Experimental Methods" section. The last official stable release is encyclopedia-2.12.30, dated December 30, 2022.

Reviewer #2

(Remarks to the Author)

In this manuscript, a workflow is introduced using a new commercial mass spectrometer, the hybrid quadrupole-LIT from Thermo. Overall, the capability of the system is relatively characterized comprehensively for proteomics study. However, there are several issues that make it hard for my supporting its publication in NC.

The manuscript serves as an application note for promoting a commercial product, which might not be appropriate for scientific journals of high impacts.

The analytical performance using the instrument and the workflow, including 50 peptides/100 ng, three orders of magnitude in calibration and etc., is good, but not stunning.

The demonstrated method works well for the Q-LIT; however, it is not clearly demonstrated why this is unique for using Q-LIT and why the results could not be obtained by a Q-Trap or Q-TOF using similar methods.

(Remarks on code availability)

Reviewer #3

(Remarks to the Author)

Rapid assay development for low-input targeted proteomics using a versatile linear ion trap

The paper is well written, easy to read and details an important area of interest to scientific readers in the proteomics area. The results and experimental model systems highlight the capability of the mass spectrometer/ LC-MS system to provide quantitative information on peptides from low amounts of material and then highlight this further with the study of immune system study of CD4+ and CD8+ cell populations, comparing to flow cytometry.

My main points of concern relate to the technical description of the quantitative proteomics area, alternative technologies and the role that this LC-MS systems can and should play in the field. For example, in the abstract (page 2) it defines triple quadrupoles as providing sensitive low-mass accuracy measurements. While this is true in how they are operated in a proteomics experiment, the mass filter can provide accurate mass values of <5ppm. The lack of resolution of the mass filter is not mentioned. These two combined give rise to the challenge, which is a lack of specificity. As such I believe a more detailed, careful and thorough discussion of the technology aspects and comparisons is required. It is also important and of note that as shown in figure 4, the quantitative measurement of peptides in matrix goes non-linear at lower levels with a LLOD being significantly lower than the LLOQ. This also occurs on triple quadrupoles, and yet is not fully explained. On higher resolution (PRM like) measurements there is a difference but this is smaller. It should be explained more thoroughly as this is interesting data and results.

Here are also some minor points to address-

Page 1- line 23- change low-mass accuracy to low specificity

Page 2- line35- change democratizing to expanding

Page 3- line 50- change to high-mass accuracy instrument "are used" rather than "required".

Page 3- line 57- extremely quick. Please add detail on how quick e.g. amu/sec or SRM transitions per sec

Page 4- line 85 low-input proteomics- what does this mean- defined for the reader

Page 5- line 92- linear ion traps are built more affordably. Despite the vacuum requirements being lower, I see no evidence that the current instrument described here is cheaper than a tandem quadrupole. I think this is somewhat irrelevant and detracts from the manuscript, remove.

Page 7- Figure 1- The schematic and workflow are good and of interest. The marketing picture of the instrument twice is unnecessary and does not add any value. Replace with something more appropriate.

Page 8- would be good to know how long it takes to create the libraries and target lists of the instrument.

Page 9- the FDR that has been used/ expected for library generation was 1%. Over 60,000 peptides have been generated in HPRP DDA and GPF-DIA analysis with a common overlap of 30,000. The 1% FDR should be added called out on this figure.

Page 12- assessing quantitative accuracy- it would be useful if this was clearer from an analytical perspective. What is the precision and accuracy of the LC-MS approach across a wide dynamic range. Line 256 mentions "reasonable quantitative accuracy". Is this less than CV 5%? This section needs to be re-written and be clear as the analytical metrics (with examples shown) of the performance of the system.

Page 13- This figure attempts to address the quantitative accuracy assessing performance in a background of di-methyl labelled pooled peptides. It is not a clear figure and I would urge the authors to pick the pertinent points and refactor this figure for ease of interpretation.

Page 14- Figure 4- how is the LLOQ/ LLOD defined? It is mentioned but not appropriately defined on the figure or legend.

Page 15- line 278- 8-10 points across the peak. Is this half-height or at base? What is the peak width and does this allow for accurate quantitative measurement. Needs expanding and more detail.

Page 20- line 361- in situations where high-resolution is impractical. Not sure what those situations are? This is a throw away comment and should be removed.

Page 20- line 363- remove word democratizing. Change sentence to "This could be impactful in immuno-oncology..."

(Remarks on code availability)

Version 1:

Reviewer comments:

Reviewer #1

(Remarks to the Author)

The new data, results, discussion, and figures incorporated during this revision have not only successfully addressed all of the reviewer's concerns but have also significantly improved the manuscript. The work is now ready for publication.

(Remarks on code availability)

The specific encyclopeDIA version used in this work is now publicly available and easily reusable for the community at <https://bitbucket.org/searleb/encyclopedia/downloads/>

Reviewer #2

(Remarks to the Author)

In the revised manuscript, the authors added a comparison between the Q-LIT and an Orbitrap instrument, further confirming that the low resolution Q-LIT could have improved sensitivity.

I don't have major questions regarding the results presented or the usefulness of this instrument for target analysis in proteomics. My main concern remains as that this manuscript reports a detailed application for a commercial product; and the reported effects are somewhat expected. The reported workflow and the details in application would be useful for potential users, but some more innovation in instrumentation or method would be expected for publication in NC.

A very minor suggestion:

Spell out GPF for the first use.

(Remarks on code availability)

Reviewer #3

(Remarks to the Author)

The revised paper entitled "Rapid Assay Development for Low Input Targeted Proteomics using a Versatile Linear Ion-Trap" is clear and the authors appear to have taken many of the review comments of the previous manuscript and incorporated these into the revised text. The basic science of the publication has not altered, but has been reinforced with more detail and points of clarification.

I still have some minor comments and concerns that should be addressed before publication. These are as follows:

1. There is a lack of consistency with technology naming. To make this read like a scientific paper and less like a product advertisement, please use Q-LIT throughout. Stellar is a product name and should be kept to a singular reference point and removed from the rest of the manuscript.
2. Page 2- line 7- remove the word "needing" high mass accuracy. It is superfluous.
3. Page 2- line 15- replace the word "economical" with valuable.
4. Page 3- line 10- change to "High mass accuracy instruments have been used in nearly all cases".
5. Page 3 paragraph 2, line 7- "While tandem quadrupole instruments are extremely capable instruments that can rapidly switch between precursor and fragment pairs (typically 0.5msec)"
6. Page 3 paragraph 2, line 10- "as such, triple quadrupoles are practically limited.."
7. Page 5, line 3- "LIT's to be built with simpler vacuum requirements than higher resolution instruments..."
8. Page 5, paragraph 3, line 2 "limited mass resolution and mass accuracy"
9. Page 6, paragraph 2, line 7 " StellarTM mass spectrometer (Thermo Scientific, USA) a new hybrid Q-LIT design". Note; To make this read like a scientific paper and less like a product advertisement, please use Q-LIT throughout. Stellar is a product name and Thermo Scientific a brand and these should be kept to this singular reference point and removed from the rest of the manuscript.
10. Page 9, line 1 "6000 proteins from only 10ng of input material" please state here in the text the calculated FDR rate
11. Page 11, line 8- remove a single workday. Specify the time taken (which in this case was 7.5 hours)
12. Page 13, line 2- the explanation of phosphopeptides being better suited to DDA needs better explanation. The stochastic nature sampling more seems nonsensical. Better explanation of why this is the case. Is t lack of specificity in processing and using DIA data?

(Remarks on code availability)

We thank the reviewers for their generally positive feedback and useful comments on our manuscript titled “Rapid Assay Development for Low Input Targeted Proteomics using a Versatile Linear Ion Trap.” Addressing these comments has improved our manuscript meaningfully, and several new experiments were performed as part of this response. We have deposited all new raw data files at both Panorama and Massive using the same accession codes. Additionally, we have added a new FAQ section to the supplementary material for the manuscript that directly addresses many of these comments as part of the publication.

Common to all reviewers was the question: “What made this work unique when other existing mass spectrometers could perform DIA and PRM measurements?” This is an important point, and we have done our best to address it in this document and the manuscript. The novelty of our work is the DIA-to-PRM workflow, as well as the software tools that enable that workflow. While our approach can be applied to a variety of instruments, we believe that the Stellar MS Q-LIT is the first major instrument platform where it will gain significant usage. This is because, unlike Q-Orbitraps or Q-ToFs, the Q-LIT is designed primarily as a targeted (PRM) instrument rather than primarily as a discovery (DIA/DDA) instrument.

Spectrum libraries are central to creating targeted assays, and they are typically collected on (or predicted for) different types of instruments than the instrument where the assays are performed. This leads to a disconnect in what can be successfully targeted because background signal interferences and matrix effects change with different HPLC systems and mass spectrometers. To solve this, we propose using GPF-DIA to help “translate” predicted libraries (as with this work) or existing DDA libraries collected on other instruments. A major advantage of our approach is that it can identify peptides and fragment ions that are viable for targeting by standardizing column conditions, gradients, and instrument parameters, all rapidly performed by adding a few runs to a routine instrument setup. By linking discovery proteomics through GPF-DIA with assay generation on the same instrument platform, our workflow (and software tools) greatly speeds up the development of targeted proteomics assays from weeks/months to a single workday. The manuscript clarifies this workflow through additional text and a new **Figure 2a**:

While we demonstrate this workflow and software using the Stellar MS Q-LIT instrument, the approach is instrument agnostic and can be performed on any instrument capable of both DIA and PRM (e.g., Q-ToFs and Q-Orbitraps). In this new revision, we have further demonstrated this aspect of our approach by repeating the DIA-to-PRM matched matrix PRM experiment using the same samples on an Exploris 480 (Q-Orbitrap). This also provides a benchmarking tool for demonstrating the advantages and drawbacks of the Q-LIT platform. Most notably, we find the Q-LIT is at least an order magnitude more sensitive than the most current Q-Orbitrap based on estimates of the lower limit of quantification for assayed peptides but less effective at detecting peptides at high input levels. Additionally, we added a new DIA-based experiment comparing detection rates for the Stellar MS versus the Exploris 480 to further test the tradeoffs between sensitivity and mass accuracy. This comparison has produced several new figures: **Figure 1b, Supplemental Figures S3, S4, S5, and S6**. While the Stellar MS is not the first Q-LIT instrument available, it is the first instrument of this class where GPF-DIA (and DIA) are practical methods. For example, the 6-injection GPF-DIA scheme required for library generation would require at least 60 injections on a Sciex QTRAP 6500+ (previously the highest-end dedicated Q-LIT). The jump in technology demonstrated by the Stellar MS is what makes our approach realistically implementable for the first time on a low-resolution platform designed primarily for targeted mass spectrometry.

Response to Reviewer 1:

Reviewer #1 (Remarks to the Author):

In this manuscript, Shannon et al. present a rapid method for developing targeted proteomics assays for low-input materials using a quadrupole-ion-trap mass spectrometer. The authors outline a workflow that involves generating data-dependent acquisition (DDA) and gas-phase fractionated data-independent acquisition (DIA) libraries at low resolution. This approach is complemented by the automatic prioritization of detectable peptides associated with the proteins of interest while maintaining the number of targets per cycle to ensure reliable quantification. The utility of this workflow is effectively demonstrated by successful quantification of low-level proteins in PRM mode from just 1 ng of sample derived from stimulated T-cells.

1. The research presented in this article is executed to a high standard, which is consistent with the authors' established reputation in the field and their strong command of the analytical techniques employed. While the work showcases valuable techniques, it appears to build upon methods that have been previously described, often by the authors themselves in earlier publications and applies them within a q-LIT instrument. These techniques include the creation of libraries, algorithms for prioritizing detectable peptides, and the use of matrix-matched calibration curves for assessing quantitative capabilities. While the application of these methods to the q-LIT instrument is certainly noteworthy, a comparison of key figures of merit with other similar systems could further highlight the innovative aspects of the Stellar.

We thank the reviewer for the kind words about our work. A recent publication from the Riley lab (PMID: 38738990) provided a comprehensive review of current mass spectrometers from large vendors. We have modified this table to include models that are relevant to this work. The table compared characteristics, such as scan rate, resolution, dynamic range, and sensitivity for each instrument to highlight the comparison between the Stellar and other systems that could be used for global and/or targeted proteomics. It should be noted that this table is generated from vendor communications and marketing materials rather than independent tests. That said, we have independently confirmed these values for Stellar. This new table is now **Supplemental Table S1**:

	instrument	m/z range	scan rate	mass resolution	dynamic range	sensitivity	release year
LIT	TFS Q-LIT (Stellar)	15–4 000	140 Hz 200 000 Da/s	0.5-2 FWHM	>1 × 10e5	atto- to femtomole	2024
QqQ	SCIEX QTRAP 6500+	50–2 000 (Trap), 5–2 000 (QqQ)	20 000 Da/s (Trap), 12 000 Da/s (QqQ)	0.7 FWHM	>1 × 10e5	atto- to femtomole	2016
	SCIEX Triple Quad 7500	5–2 000	20 000 Da/s	0.5 FWHM	>1 × 10e6	atto- to femtomole	2020
	Agilent 6495D	5–3 000	18 700 Da/s	0.4–2.5 FWHM	>6 × 10e6	atto- to femtomole	2023
	Shimadzu 8060NX	2–2 000	30 000 Da/s	0.5 FWHM	1 × 10e7	atto- to femtomole	2020
	TFS TSQ Altis	5–2 000	15 000 Da/s	0.2–2.0 FWHM	>1 × 10e6	atto- to femtomole	2019
ToF	Waters Synapt XS	20–64 000	30 Hz	75 000	1 × 10e4	femtomole	2019
	Bruker timsTOF SCP	50–20 000	120 Hz	60 000	5 × 10e4	zepto- to attomole	2021
	Bruker timsTOF HT	50–20 000	150 Hz	60 000	1 × 10e5	atto- to femtomole	2022
	Bruker timsTOF Ultra	50–20 000	300 Hz	60 000	5 × 10e4	zepto- to attomole	2023
	SCIEXZenoTOF 7600	40–40 000	133 Hz	42 000	1 × 10e5	atto- to femtomole	2021
Orbitrap	TFS Orbitrap Exploris	40–8 000*	40 Hz	480 000	1 × 10e5	femtomole	2019
	TFS Orbitrap Eclipse Tribid	40–8 000	40 Hz (Orbitrap) 45 Hz (LIT)	1 000 000	>5 × 10e3	atto- to femtomole	2019
	TFS Orbitrap Ascend Tribid	40–16 000*	45 Hz (Orbitrap) 50 Hz (LIT)	1 000 000	5 × 10e3	atto- to femtomole	2022
Ast	Q-Orbitrap-Astral	40–2 000	200 Hz	80 000	>1 × 10e4	zepto- to attomole	2023

For clarity, we added the Stellar MS (yellow highlighted row), the SCIEX QTRAP 6500+ (blue highlighted row), and the Thermo Fisher Orbitrap Ascend Tribid (pink highlighted row). The QTRAP represents the previous highest-end dedicated Q-LIT platform, while the Orbitrap Ascend is the most recent tribid (quadrupole, LIT, and Orbitrap). The QTRAP is listed as a triple quadrupole since it is generally operated in this mode. However, we have annotated key characteristics that differ when the QTRAP is operated as either an ion trap or a triple quadrupole.

2. For instance, how does the Stellar compare to the Orbitrap Ascend when operated solely with the quadrupole and linear ion trap?

As shown in the table, the Q-LIT instrument is capable of scanning more quickly than most previous Thermo Fisher Scientific (TFS) mass spectrometers. Relative to the Orbitrap Ascend, the LIT in the Stellar scans 2.8× faster than the LIT in the Ascend. In addition, the ion optics are somewhat improved due to the more direct ion routing from removing components necessary for Tribrid-specific functions (**Figure 1**). We have found that this results in an increase in the number of potential targets in PRM assays and higher detection performance using DIA. While not a direct comparison, HeLa DIA peptide detections in LIT-only mode with the Orbitrap Eclipse from our paper Phlairaharn et al (2023) show marked improvement both in the software tools and instrument performance between generations, resulting in >2x detections at all input levels:

Please note that the “Orbitrap” data is not comparable since in Phlairaharn et al we operated the Eclipse in Orbi/LIT mode, while in this work, we operated an Exploris 480 in Orbi/Orbi mode for MS1/MS2 measurements.

We have also added this text to **Supplemental Note 1** Frequently Asked Questions (#1), pointing readers to Remes et al (2024) for a more detailed discussion of this question:

“How does the Stellar compare to other unit-resolution instruments?”

As shown in supplemental table S7, the Q-LIT instrument is capable of scanning more quickly compared to most previous Thermo Fisher Scientific (TFS) mass spectrometers, including the Q-Orbitrap LIT Ascend. Additionally, in Remes et al paper titled “Hybrid Quadrupole Mass Filter – Radial Ejection Linear Ion Trap and Intelligent Data Acquisition Enable Highly Multiplex Targeted Proteomics,” the authors compare figures of merit (limit of detection and limit of quantification) for the TFS TSQ Altis (QqQ analyzers) to the Stellar (Q-LIT) MS system. Specifically, in Figure 3 of Remes’ work, they show the differences in detection for the Orbitrap Astral to the Stellar and the %CV of Astral vs Stellar PRM assays to show their similarity.”

3. Additionally, how does it perform against other low-resolution instruments capable of PRM-like measurements from different vendors, or even against available high-resolution instruments from the same vendor? Addressing this aspect could significantly enhance the manuscript's impact and innovation aspect. Therefore, while I appreciate the quality of the research, I would suggest further exploration of these comparisons before considering publication in Nature Communications.

We thank the reviewer for pointing out this critical direction. In this revision, we have benchmarked the Q-LIT platform relative to a Q-Orbitrap (Exploris 480) at both DIA and PRM. Across these experiments, we found the Q-LIT is approximately an order of magnitude more sensitive than the Q-Orbitrap but less effective at detecting peptides at high input levels. Furthermore, at high input, different quantitative characteristics around handling high dynamic range measurements may make one platform more appealing than the other for specific

experiments. This comparison has produced several new figures: **Figure 1b, Supplemental Figures S3, S4, S5, and S6.**

As mentioned above, we have also included the new **Supplemental Table S1** comparing Stellar to other vendor platforms, including both high- and low-resolution instruments. The following paragraph has been added to page 7 of the manuscript:

*“The Stellar Q-LIT platform is a versatile instrument that can perform both global (DIA) and targeted (PRM) proteomics with the same instrument. Dedicated low-resolution triple quadrupole instruments are capable of highly sensitive measurements with wide dynamic ranges. However, they are limited to selected reaction monitoring (SRM) for specific precursor/fragment ion transitions. The Sciex QTRAP platform is a hybrid triple quadrupole that can scan as an ion trap in the last quad, making it also capable of PRM and DDA. While similar in geometry to the Stellar, the QTRAP is not configurable to perform DIA, in part because of its slower speed. In contrast, Stellar can scan approximately 10× faster than the QTRAP 6500+, making it an interesting candidate low-resolution instrument for developing a stand-alone workflow for transitioning global results to targeted assays. Other instruments are also capable of performing both global and targeted scans; therefore, we wanted to compare the Q-LIT to existing instrumentation using the Q-Orbitrap (Exploris 480) as a benchmark. Performance characteristics of other related instruments from a wide variety of vendors are tabulated in **Supplemental Table S1**, which was modified from a recent literature review by Peters-Clarke et al.”*

Finally, we have performed additional experiments where we have compared the sensitivity of the Q-LIT platform to a Q-Orbitrap (Exploris 480). This comparison has produced several new figures: **Figure 1b-c, Supplemental Figures S3, S4, S5, and S6** have now been created or updated to include comparisons of Q-LIT and Q-Orbitrap data. This new data tests several aspects, including LoDs/LoQs (**S3**), and LoD and LoQ ratios between instruments (**S4**), PRM accuracy (**S5**), PRM points across the peak (**S6**), and detection rates in DIA at different input levels (**Figure 1b**).

Reviewer #1 (Remarks on code availability):

4. There is no easily identifiable release on the main page corresponding to the version v3.0.0-SNAPSHOT mentioned in the "Experimental Methods" section. The last official stable release is encyclopedia-2.12.30, dated December 30, 2022.

The build of EncyclopeDIA used in this manuscript is now tagged “EncyclopeDIA v-4.7.11”. We have released the binary (encyclopedia-4.7.11-executable.jar) for download on the EncyclopeDIA website and made it available on the MASSIVE/Panorama repositories. However, it has some incomplete features (unrelated to this work), so it is not listed as a “stable” release.

We have also added this text to **Supplemental Note 1** Frequently Asked Questions (#5):

“What version of EncyclopeDIA should be used to process ion trap data for sensitive quantification?
A tagged version of EncyclopeDIA titled “encyclopedia-4.7.11-executable.jar” is the best version to use for IonTrap/IonTrap data. This version can be found in the Raw Files on the panorama public page for this work, or on EncyclopeDIA’s bitbucket, under tagged versions in the source code. Here is a direct URL:
<https://bitbucket.org/searleb/encyclopedia/downloads/encyclopedia-4.7.11-executable.jar>”

Response to Reviewer 2:

Reviewer #2 (Remarks to the Author):

In this manuscript, a workflow is introduced using a new commercial mass spectrometer, the hybrid quadrupole-LIT from Thermo. Overall, the capability of the system is relatively characterized comprehensively for proteomics study. However, there are several issues that make it hard for my supporting its publication in NC.

1. The manuscript serves as an application note for promoting a commercial product, which might not be appropriate for scientific journals of high impact.

We kindly thank the reviewer for their feedback regarding the potential conflict of interest posed by collaborating with industry scientists. These concerns have merit, but we respectfully disagree that this work functions as an application note for Stellar. As stated above, the novelty of our work is less in the specific DIA and PRM instrumentation than in the DIA-to-PRM workflow and software tools that enable that workflow. To emphasize this, we have removed all “branding” images of the Stellar from the main figures in this revision.

We note that Stellar is not the only platform capable of this workflow and clarify that with several new experiments comparing the Stellar to a high-resolution Q-Orbitrap (the Exploris 480). This comparison has produced several new figures: **Figure 1b**, **Supplemental Figures S3, S4, S5, and S6**. Other instrument types have the ability to collect both DIA and PRM, and we have included a breakdown of the most current instruments from all major instrument vendors in **Supplemental Table S1**. However, low-resolution Q-LITs stand alone with their ability to combine global and targeted measurements in a cost-efficient platform. Previous Q-LIT instruments (e.g., several models of the SCIEX QTRAP) are at least an order of magnitude slower than the Stellar, making DIA effectively impossible. As far as we are aware, SCIEX QTRAPs are also software-locked so as to not allow DIA acquisition. As such, this work was only made possible with the development of the Stellar MS.

We identified several strengths and weaknesses of the Stellar MS to existing high-resolution mass spectrometers using the most recent generation Q-Orbitrap as a benchmark. First, we found that the Q-LIT has similar quantitative accuracy to the Q-Orbitrap but with less bias towards 0 in low-intensity measurements. Matching cycle time between the instruments, we observed that the Q-LIT was also nearly an order of magnitude more sensitive for most measured peptides. At low input, the Q-LIT could measure more peptides with PRM and exhibited lower LoQ and LoD values. However, at high-input levels (>50 ng on column), the Q-Orbitrap generally outperformed the Q-LIT in terms of PRM dynamic range and number of DIA detections (discussed below).

The PRM sensitivity result is shown in **Supplemental Figure S3**, where we measured the same peptides using both platforms:

The **Supplemental Figure S3** legend notes: “This figure emphasizes the types of quantitative errors observed by both instruments. The Q-LIT tends to measure higher values where there are small signals (as indicated by the box plots shifting up at lower dilution ratios) due to higher interference caused by lower resolution. In contrast, the Q-Orbitrap tends to measure 0 when there are small signals (as indicated by the box plots shifting down at lower dilution ratios) due to the limited capacity of the Orbitrap.”

Additionally, **Supplemental Figure S4** shows that the Stellar typically produces lower LoQ and LoD values:

Here, we found that signal dropout in the Orbitrap causes “modes” where LoD or LoQ values are forced to fit because only zeroes are reported below that point. In contrast, the background noise matches closer to the expectations of common LoD and LoQ fitting algorithms, and thus, those values are typically better estimated. Echoing previous findings in our lab, we observed this sensitivity with DIA, where we detected >6,000 proteins from 10 ng of HeLa digest. However, the Q-Orbitrap equaled or outperformed the Q-LIT at sufficiently high input (50 ng or above). We believe this is a limitation of trading off mass accuracy for narrow precursor isolation, where better resolution is preferable when measuring a higher number of potential peptides and proteins. This result is now shown in **Figure 1b**, reorganized here for clarity:

2. The analytical performance using the instrument and the workflow, including 50 peptides/100 ng, three orders of magnitude in calibration and etc., is good, but not stunning.

The goal of our work is not to quantify the highest number of peptides but to quantify a substantial number of biologically relevant proteins, acting as a companion to flow cytometry for monitoring specific proteins in rare cell populations. In the new Q-Orbitrap comparison (above), we found that the Q-LIT could schedule more peptides than the Q-Orbitrap without compromising the windows used for scheduling. For the 100 ng PRM assay, we quantified 473 of 541 targeted peptides in at least two dilutions, all of which are biologically relevant to T cell function. In contrast, with the Q-Orbitrap we could only schedule 300 targets when trying to maintain the same number of points across the peak. For the 1 ng PRM assay, we quantified 61 of 102 target peptides in at least two dilutions and only 19 of 102 with the Q-Orbitrap in a parallel experiment.

We agree that it is possible to target far more peptides using high input (1 ug) and advanced scheduling using Adaptive RT scheduling (e.g., PMID: 32867497, as demonstrated with the Stellar MS in this preprint PMID: 38895256). However, the focus of this work is to develop techniques to target fewer peptides at low input with high fidelity. Furthermore, we were interested in quantifying specific, difficult-to-measure proteins, such as kinases, transcription factors, and immune markers. As a result, we tuned our instrument parameters to more aggressively measure low-input signals by increasing ion injection times and lengthening scan times. To be clear, these parameters are not ideal for general proteomics approaches of high-input samples (e.g., >100 ng) measuring high-abundant proteins. For generic proteomics measurements of high-input samples, we suggest using methods from the preprint PMID: 38895256.

When using DIA methods, the new **Figure 1b** shows Stellar produced >2x more peptide detections at all input levels in comparison to the Eclipse when operated in Ion trap/Ion trap mode (as we previously published in PMID: 37338819). This comparison is shown again for clarity:

[Editorial note: this figure was redacted due to third-party rights. It can be found in Phlairaharn et al., 2023, Figure 2a]

This work, **Figure 1b**:

Using these approaches, we detected >7,000 peptides (>3,000 proteins) from only 1 ng of HeLa with a unit-resolution ion trap.

3. The demonstrated method works well for the Q-LIT; however, it is not clearly demonstrated why this is unique for using Q-LIT and why the results could not be obtained by a Q-Trap or Q-TOF using similar methods.

While these methods could not be realistically performed on the current Q-Trap 6500+ due to the slow speed of that instrument (approximately 10x slower than the Stellar MS), they could be performed on Q-Exactives and Q-ToFs. We have included a further breakdown of Stellar compared to different instruments from other vendors in the new **Supplementary Table S1**. As discussed above, with this new revision, we now demonstrate the same methods with a top-of-the-line Q-Orbitrap (Exploris 480) and provide a thorough comparison between the approaches.

We believe these new experiments (shown in **Figure 1b**, **Supplemental Figures S3, S4, S5, and S6** above) demonstrate that our rapid assay development workflow is not tied to any one instrument and that the strengths of the Stellar platform at low input loads are balanced with weaknesses at higher input. We believe

Orbitraps are more performant instruments (albeit at a significantly higher cost) when working with high-input samples. However, we believe that the Stellar should outperform Orbitraps at 10 ng or lower for both DIA and PRM, even with only unit resolution.

Response to Reviewer 3:

Reviewer #3 (Remarks to the Author):

Rapid assay development for low-input targeted proteomics using a versatile linear ion trap

The paper is well written, easy to read and details an important area of interest to scientific readers in the proteomics area. The results and experimental model systems highlight the capability of the mass spectrometer/ LC-MS system to provide quantitative information on peptides from low amounts of material and then highlight this further with the study of immune system study of CD4+ and CD8+ cell populations, comparing to flow cytometry.

1. My main points of concern relate to the technical description of the quantitative proteomics area, alternative technologies and the role that this LC-MS systems can and should play in the field. For example, in the abstract (page 2) it defines triple quadrupoles as providing sensitive low-mass accuracy measurements. While this is true in how they are operated in a proteomics experiment, the mass filter can provide accurate mass values of <5ppm. The lack of resolution of the mass filter is not mentioned. These two combined give rise to the challenge, which is a lack of specificity. As such I believe a more detailed, careful and thorough discussion of the technology aspects and comparisons is required.

We thank the reviewer for suggesting that we include more specifications on the quadrupole. While the boundaries of modern quadrupole filters can be precisely set, we are not aware of any quadrupole that can provide meaningful ion transmission with a <5 ppm width window. In this work, we operate the quadrupole using the following precursor isolation widths, which are generally greater than the 0.2 to 2 m/z FWHM resolution of Q₁:

Method	Isolation Window
PRM	2.0 m/z
GPF-DIA	2.0 m/z
DIA	8.0 m/z
DDA	1.6 m/z

We have added new details on the quadrupoles in the manuscript on page 7:

“The instrument shares many of the same design components as existing Orbitrap-based instruments.^{27–29} Ions are passed through a mass filter quadrupole (Q₁), then concentrated within the ion routing multipole (Q₂) before mass analysis in the LIT. The Q₁ mass filter upstream of the LIT is designed to increase ion transmission using an optimized rod shape with hyperbolic surfaces that allow for isolation windows as small as 0.2 to 2 m/z FWHM (typically below 1 m/z).³⁰ Within this work, we generally maintain a minimum of 2 m/z isolation windows for PRM to capture multiple isotopes per precursor simultaneously, thus increasing sensitivity. These configurations produce high scanning speeds by performing ion accumulation in parallel with mass analysis in the low-pressure LIT.^{31,32}”

The resolution of the linear ion trap is 0.5 to 2 m/z FWHM. Overall, the Stellar platform as a whole has similar specificity to most commercial triple quadrupoles, now noted in **Supplemental Table S1**. Also, since Q₁ is only minorly updated from current Thermo Fisher hybrids and tribrids, it exhibits very similar performance to those existing instruments.

We have also added the following discussion about Q₁ to the **Supplemental Note 1** Frequently Asked Questions (#4):

“How does the QR5 Plus quadrupole mass filter improve ion transmission on the Stellar?”

The mass filter, located upstream of the LIT, is designed to increase ion transmission using a 5.25 mm field radius device. The field radius is the radial distance from the center of the quadrupole where the field strength

is stable and uniform for ion transmission. When the field radius increases, the range of stable ion trajectories is broadened, allowing for higher ion transmission. Decreasing the field radius size restricts the paths ions can take, consequently decreasing ion transmission. The 5.25 mm field radius is an improvement over the 4.0 mm field radius of the LIT in the Velos Pro and other modern Thermo ion traps.”

2. It is also important and of note that as shown in figure 4, the quantitative measurement of peptides in matrix goes non-linear at lower levels with a LLOD being significantly lower than the LLOQ. This also occurs on triple quadrupoles, and yet is not fully explained. On higher resolution (PRM like) measurements there is a difference but this is smaller. It should be explained more thoroughly as this is interesting data and results.

We thank the reviewer for suggesting this additional discussion. We have added the following text on page 15: *“As the analyte signal dropped with decreasing concentration, background interference tended to overwhelm the analyte signal. As a result, quantitative ratios with the Q-LIT tend to regress to 1:1, resulting in higher than expected measured ratios.”*

In addition to the new Q-Orbitrap parallel experiment (discussed both above and in more detail below), we have also added the following text pertaining to how LITs are different from Orbitraps on page 15:

“Q-Orbitraps have limited trapping capacity, which can limit the dynamic range within a spectrum. In contrast to the Q-LIT, peptides with low background signal simply stop being measured before they fall below the LoQ, resulting in quantitative ratios regressing to 0:1, resulting in lower than expected measured ratios (Supplemental Figure S3). A consequence of this signal drop-off is that estimated LoQ and LoD values for the Q-Orbitrap can be more difficult to estimate correctly, producing modes around samples where the peptide signal falls below the spectrum dynamic range (Supplemental Figure S4). Although the distribution of LoDs remains similar between the Q-LIT and Q-Orbitrap, the distribution of LoQs is typically lower for the Q-LIT on a peptide-by-peptide basis (Supplemental Figure S5).”

Finally, we have included a deeper discussion of this as part of **Supplemental Note 1** Frequently Asked Questions (#3):

“What are the differences between the LoD and LoQ obtained from a high-resolution instrument compared to a low-resolution instrument?”

As the analyte signal drops with decreasing concentration, background interference can overwhelm the analyte signal. The point at which that background signal appears is referred to as the LoQ, where a change in signal directly reflects a change in analyte quantity. The point at which the background signal eclipses the analyte signal is the LoD, below which the analyte is not detectable. When quantitative ratios are calculated, the lowest signal drops below the LoQ first, generally resulting in a regression towards 1:1. With regards to peptide calibration curves, the point at which the ratio becomes nonlinear indicates the LoQ.

Instruments with low resolving power (LITs and quadrupoles) show more interference than instruments that can resolve small mass differences between ions, resulting in more background signal. In contrast, Orbitraps have limited trapping capacity, which limits the dynamic range within a spectrum. In some cases, some peptides with low background signal simply stop being measured before they fall below the LoQ. In this case, the LoQ can be hard to estimate since it is essentially below the LoD.”

3. Here are also some minor points to address-
Page 1- line 23- change low-mass accuracy to low specificity

We have made this change to the abstract.

4. Page 2- line35- change democratizing to expanding

We have made this change to the abstract.

5. Page 3- line 50- change to high-mass accuracy instrument “are used” rather than “required”.

We have made this change on page 3.

6. Page 3- line 57- extremely quick. Please add detail on how quick e.g. amu/sec or SRM transitions per sec

We have added the text (bold):

“While triple quadrupoles are extremely quick instruments capable of rapidly switching between ion pairs (typically 0.5 msec dwell time), they can only monitor a single m/z at a time, which makes generating even low-resolution full spectra impractical.”

7. Page 4- line 85 low-input proteomics- what does this mean- defined for th reader

On page 4, we define low-input proteomics as being at or below 10 ng for high-resolution mass analyzers and higher-input samples as being at or above 100 ng.

8. Page 5- line 92- linear ion traps are built more affordably. Despite the vaccum requirements being lower, I see no evidence that the current instrument described here is cheaper than a tandem quadrupole. I think this is somewhat irrelevant and detracts from the manuscript, remove.

This sentence was intended to discuss vacuum requirements and costs in reference to higher-resolution instruments like Orbitraps or ToFs. We have more explicitly clarified this and eliminated the affordability discussion in this section. The sentence on page 5 now reads:

“Lower vacuum pump requirements allow LITs to be built more robustly than higher resolution instruments, such as Orbitrap or ToF analyzers, and housed in smaller instrument footprints.”

9. Page 7- Figure 1- The schematic and workflow are good and of interest. The marketing picture of the instrument twice is unnecessary and does not add any value. Replace with something more appropriate.

We thank the reviewer for indicating this deficiency and have modified the figure to eliminate the marketing picture entirely. The workflow subpanel from **Figure 1** has been moved to **Figure 2a** to make room for more instrument characterization in **Figure 1** and focus on more relevant details about the scheduling algorithm. **Figure 2a** now looks like:

10. Page 8- would be good to know how long it takes to create the libraries and target lists of the instrument.

The following text has been added to the manuscript on page 11:

“The entire workflow can be performed in a single workday, from translation library to PRM assay. For this work, a single library was generated in 7.5 hours, including 6× hour-long gradients followed by 15 minutes for

sample loading and column equilibration, where the translation library was processed in parallel with the acquisition.”

11. Page 9- the FDR that has been used/ expected for library generation was 1%. Over 60,000 peptides have been generated in HPRP DDA and GPF-DIA analysis with a common overlap of 30,000. The 1% FDR should be added called out on this figure.

We have made this change to the legend for **Figure 2**.

12. Page 12- assessing quantitative accuracy- it would be useful if this was clearer from an analytical perspective. What is the precision and accuracy of the LC-MS approach across a wide dynamic range.

We thank the reviewer and have performed a new experiment to assess the performance of peptides in a HeLa proteome over a wide dynamic range. To accomplish this, we spiked 15 heavy labeled Pierce Retention Time Calibration (PRTC) peptides at a ratio of 100 fmol PRTC to 100 ng HeLa. We performed PRM experiments to measure these peptides, showing the precision and accuracy of the instrument with water dilutions over >5 orders of magnitude. Of the 15 heavy labeled peptides, we regularly only observe 14 with our HPLC setup, and the Stellar showed that same behavior. We observed high quantitative accuracy over 4 orders of magnitude for most of the 14 peptides, and some peptides are quantitative to 5 orders of magnitude, matching the vendor specifications in **Supplemental Table 1**. Below 5 orders of magnitude we found that the peptide signals were polluted with some background noise, raising them off the expected dashed line. These results are now shown in **Figure 1c**:

13. Line 256 mentions “reasonable quantitative accuracy”. Is this less than CV 5%? This section needs to be re-written and be clear as the analytical metrics (with examples shown) of the performance of the system.

We thank the reviewer for bringing this to our attention. We have added a new **Supplemental Figure S2** to address accuracy in terms of absolute error from the expected ratios. While these errors can be high relative to other metrics, they are more representative of true quantitative accuracy than % CVs, which simply measure reproducibility. The absolute error levels for the Q-LIT matrix-match calibration curve at (a) 100 ng, (b) 10 ng, and (c) 1 ng of material on-column are typically within an acceptable absolute error rate of $\pm 50\%$ absolute error on the estimated ratio (pink band) out to 1/20, but accuracy falls off after 1/100. Note that the acceptable error band shrinks as target ratios become smaller, making it harder to hit precisely. For example, the pink band at 10% indicates 5% to 15%, while the pink band at 1% indicates 0.5% to 1.5%. (d) Shows the percent of peptides within the pink band at a target ratio:

We do calculate % CVs for the 1 ng assay in **Figure 6**. We have updated Figure 6's legend to clarify that we are assessing quantitative accuracy by repeated measurements. In this figure, we use a % CV of <20% to indicate reproducible quantitative accuracy. The legend now states:

"Coefficient of technical variation (% CV) plots for all peptides quantified in the 1 ng assay. The red dotted line indicates 20% CV on each plot."

14. Page 13- This figure attempts to address the quantitative accuracy assessing performance in a background of di-methyl labelled pooled peptides. It is not a clear figure and I would urge the authors to pick the pertinent points and refactor this figure for ease of interpretation.

We agree that **Figure 3** was too complex and tried to tackle too many concepts. In the main figure have retained the the scatter plots to the left and moved the LoD/LoQ histograms to **Supplemental Figure S5**, which is focused exclusively on LoD/LoQ distributions compared between the Q-LIT and the Q-Orbitrap. We have opted to retain a simplified interpretation of the LoD/LoQ estimates as boxplots (**Figures 3e** and **3f**). We thank the reviewer for helping us find this more intuitive visualization:

15. Page 14- Figure 4- how is the LLOQ/ LLOD defined? It is mentioned but not appropriately defined on the figure or legend.

EncyclopeDIA uses a common textbook definition of LoD and LoQ. The legend for **Figure 4** now reads: *“LoQ and LoD were estimated on a peptide-by-peptide basis using EncyclopeDIA’s curve fitting algorithm. First, the algorithm determines a maximum line through the noise of the calibration curve and then fits the linear dynamic range. The intersection of both lines is the LoD, while the LoQ is three standard deviations of the noise above the LoD.”*

16. Page 15- line 278- 8-10 points across the peak. Is this half-height or at base? What is the peak width and does this allow for accurate quantitative measurement. Needs expanding and more detail.

We have simplified this discussion to characteristics of the actual assay and more fully developed this in the new **Supplemental Figure S6**. On page 18 we report: *“All told, 61 peptides with estimable LoQs in the 1 ng assay corresponded to 30 quantified proteins with a median of 11-14 points across the peak base using the Q-Orbitrap and Q-LIT (Supplemental Figure S6).”*

Supplemental Figure S6 shows underlying distributions of peptide peak widths (in seconds) and the number of points across the peak base (as calculated by Skyline) for the 100, 10, and 1 ng assays on both the Q-LIT and the Q-Orbitrap. The peak width distributions are consistent across the Q-Orbitrap (10 ng and 100 ng) and Q-LIT (all assays), ranging from 18.6-21.6 seconds for the median peak width. At 1 ng on the Q-Orbitrap, peak widths were a median of 12 seconds, which is likely due to the low signal observed at that amount with the Orbitrap analyzer. As this figure demonstrates, we adjusted the Q-Orbitrap assay parameters to match the Q-LIT in points across the peak base as closely as possible, although there are some discrepancies (in particular, the Orbitrap 1 ng assay):

a Points Across the Peak

b Peak Widths

17. Page 20- line 361- in situations where high-resolution is impractical. Not sure what those situations are? This is a throw away comment and should be removed.

We have cut this sentence from the manuscript.

18. Page 20- line 363- remove word democratizing. Change sentence to “This could be impactful in immuno-oncology....”

We have removed this phrasing. The section now reads, “*We believe these instruments offer a high value-to-expense ratio, potentially providing more affordable mass spectrometry instrumentation on a broader array of laboratory settings.*” on page 24.

We thank the reviewers for their helpful comments. We agreed with all suggestions given, and have changed the text to reflect this. Most changes were centered around using consistent naming with the Q-LIT platform and limiting usage of the commercial brand name (Stellar).

Response to Reviewer 1

The new data, results, discussion, and figures incorporated during this revision have not only successfully addressed all of the reviewer's concerns but have also significantly improved the manuscript. The work is now ready for publication.

Reviewer #1 (Remarks on code availability):

The specific encyclopeDIA version used in this work is now publicly available and easily reusable for the community at <https://bitbucket.org/searle/encyclopedia/downloads/>

We also included additional information to help readers find the version of EncyclopeDIA that was used in this work. We now point out the specific version under "Data/code availability" and direct users to the downloads page for the bitbucket website.

Response to Reviewer 2

In the revised manuscript, the authors added a comparison between the Q-LIT and an Orbitrap instrument, further confirming that the low resolution Q-LIT could have improved sensitivity.

I don't have major questions regarding the results presented or the usefulness of this instrument for target analysis in proteomics. My main concern remains as that this manuscript reports a detailed application for a commercial product; and the reported effects are somewhat expected. The reported workflow and the details in application would be useful for potential users, but some more innovation in instrumentation or method would be expected for publication in NC.

A very minor suggestion:x

Spell out GPF for the first use.

Fixed on page 9. The sentence now reads: "We implemented a workflow and software tool to take advantage of the ability to generate peptide libraries using off-line fractionated DDA or gas-phase fractionation data-independent acquisition (GPF-DIA) and developed software to build on-the-fly PRM assays for the same instrument. (**Figure 2a**)."

Response to Reviewer 3

The revised paper entitled "Rapid Assay Development for Low Input Targeted Proteomics using a Versatile Linear Ion-Trap" is clear and the authors appear to have taken many of the review comments of the previous manuscript and incorporated these into the revised text. The basic science of the publication has not altered, but has been reinforced with more detail and points of clarification.

I still have some minor comments and concerns that should be addressed before publication. These are as follows:

1. There is a lack of consistency with technology naming. To make this read like a scientific paper and less like a product advertisement, please use Q-LIT throughout. Stellar is a product name and should be kept to a singular reference point and removed from the rest of the manuscript.

Thank you for pointing this out. We have changed the instrument name from “Stellar” to “Q-LIT” in most places where it appears in the manuscript. The one place where we keep the name “Stellar” is when comparing the QTRAP from Sciex to the Stellar from Thermo, where we were specifically discussing features of each instrument model on page 6.

2. Page 2- line 7- remove the word “needing” high mass accuracy. It is superfluous.

This word has been removed.

3. Page 2- line 15-replace the word “economical” with valuable.

The word has been replaced in this sentence.

4. Page 3- line 10- change to “High mass accuracy instruments have been used in nearly all cases”.

The sentence has been changed.

5. Page 3 paragraph 2, line 7- “While tandem quadrupole instruments are extremely capable instruments that can rapidly switch between precursor and fragment pairs (typically 0.5msec)”

The text has been changed to clarify the sentence.

6. Page 3 paragraph 2, line 10- “as such, triple quadrupoles are practically limited..”

The word practically has been added.

7. Page 5, line 3- “LIT’s to be built with simpler vacuum requirements than higher resolution instruments...”

Thank you for this comment, we agree that this enhances our meaning. The text has been changed.

8. Page 5, paragraph 3, line 2 “limited mass resolution and mass accuracy”

The words “mass accuracy” have been added to the end of this sentence.

9. Page 6, paragraph 2, line 7 “ Stellar™ mass spectrometer (Thermo Scientific, USA) a new hybrid Q-LIT design”. Note; To make this read like a scientific paper and less like a product advertisement, please use Q-LIT throughout. Stellar is a product name and Thermo Scientific a brand and these should be kept to this singular reference point and removed from the rest of the manuscript.

We thank you for noting this important aspect of the manuscript. We now refer to the “Stellar” as a “Q-LIT” in all portions of the manuscript, except in places where model-specific details are discussed to maintain clarity within the text. In particular, the section referenced on page 6, paragraph 2 has been changed to remove mention of the brand name.

10. Page 9, line 1 “6000 proteins from only 10ng of input material” please state here in the text the calculated FDR rate.

We have specified in this sentence that the results are filtered to a 1% FDR rate.

11. Page 11, line 8- remove a single workday. Specify the time taken (which in this case was 7.5 hours)

The sentences have been altered to reflect this.

12. Page 13, line 2- the explanation of phosphopeptides being better suited to DDA needs better explanation. The stochastic nature sampling more seems nonsensical. Better explanation of why this is the case. Is it a lack of specificity in processing and using DIA data?

DDA and DIA produce different behaviors when analyzing phosphopeptide positional isomers.¹ With DIA, positional isomer identification is biased based on abundance and typically does not change run-to-run. In contrast, with DDA, dynamic exclusion, meant to favor low-abundance peptide detection, can inadvertently change which isomers are detected. As phosphopeptides elute, the first isomer can place the shared precursor mass on an exclusion list, and the second isomer (which may actually be more abundant) could be skipped altogether. Additionally, DDA's intensity sampling bias can sometimes miss the first isomer if it is too low in abundance, allowing the second to be targeted instead. Taken together, these factors reduce the consistency of identifying phosphopeptide isomers compared to non-modified peptides in DDA experiments.² While this is a clear challenge for quantitative experiments, in the case of library generation, this stochasticity produces an unexpected advantage by increasing the coverage of different positional isomers across technical replicates.

To clarify this point in the text without detracting from the main message of the paper, we have changed the sentence to include an abbreviated explanation of this as follows: "However, some sample types, such as enriched phosphopeptides, may be better suited to library generation with DDA since the semi-stochastic sampling of precursor ions for fragmentation allows for a greater number of unique positional isomers to be detected when combining technical replicates."

References

1. Joyce, A. W. & Searle, B. C. Computational approaches to identify sites of phosphorylation. *Proteomics* **24**, e2300088 (2024).
2. Wolf-Yadlin, A., Hautaniemi, S., Lauffenburger, D. A. & White, F. M. Multiple reaction monitoring for robust quantitative proteomic analysis of cellular signaling networks. *Proc. Natl. Acad. Sci. U. S. A.* **104**, 5860–5865 (2007).